# Mechanistic insights into $CO_2$ conversion chemistry of copper bis-(terpyridine) molecular electrocatalyst using accessible operando spectrochemistry

Huihui Zhang[1,3], Chang Xu[1,2,3], Xiaowen Zhan[1] ✉, Yu Yu[1], Kaifu Zhang[1], Qiquan Luo [1] ✉, Shan Gao[1] ✉, Jinlong Yang [2] & Yi Xie [2] ✉

The implementation of low-cost transition-metal complexes in $CO_2$ reduction reaction ($CO_2RR$) is hampered by poor mechanistic understanding. Herein, a carbon-supported copper bis-(terpyridine) complex enabling facile kilogram-scale production of the catalyst is developed. We directly observe an intriguing baton-relay-like mechanism of active sites transfer by employing a widely accessible operando Raman/Fourier-transform infrared spectroscopy analysis coupled with density functional theory computations. Our analyses reveal that the first protonation step involves Cu-N bond breakage before the *COOH intermediate forms exclusively at the central N site, followed by an N-to-Cu active site transfer. This unique active site transfer features energetically favorable *CO formation on Cu sites, low-barrier CO desorption and reversible catalyst regeneration, endowing the catalyst with a CO selectively of 99.5 %, 80 h stability, and a turn-over efficiency of 9.4 s$^{-1}$ at −0.6 V vs. the reversible hydrogen electrode in an H-type cell configuration. We expect that the approach and findings presented here may accelerate future mechanistic studies of next-generation $CO_2RR$ electrocatalysts.

Harvesting intermittent energy such as solar and wind power in the form of "renewable fuels" is indispensable for future energy systems. In this context, the electrochemical reduction of $CO_2$ driven by renewable electricity is one attractive approach to convert $CO_2$ into valuable fuels and feedstocks[1,2]. Nonetheless, the rational design of highly efficient electrocatalysts for $CO_2$ reduction reaction ($CO_2RR$) is still largely hampered by poor mechanistic understanding[3]. In-situ/*operando* spectroscopic techniques can provide key complementary information by investigating electrocatalysis under electrochemical reactions[4–7]. In particular, Raman spectroscopy have the potential to

provide in-depth insights into the complex reaction pathways by capturing the reaction intermediates and identifying the active sites during reaction processes[8,9]. In an innovative fashion, Tian et al. unprecedentedly demonstrated that the in-situ surface-enhanced Raman spectroscopy (SERS) technique offers an effective and reliable way to probe real-time intermediates formation and detect reaction pathways for the oxygen reduction reaction (ORR)[10]. Compared with oxygen evolution reaction and ORR, $CO_2RR$ is a more complicated process involving proton coupled multi-electron transfer. However, we noticed that thus far common in-situ/*operando* Raman

[1]School of Chemistry and Chemical Engineering, School of Materials Science and Engineering, Institute of Physical Science and Information Technology, Anhui Province Key Laboratory of Chemistry for Inorganic/Organic Hybrid Functionalized Materials, Key Laboratory of Structure and Functional Regulation of Hybrid Materials of Ministry of Education, Anhui University, Hefei 230601 Anhui, P. R. China. [2]Hefei National Laboratory for Physical Sciences at Microscale, University of Science and Technology of China, Hefei 230026 Anhui, P. R. China. [3]These authors contributed equally: Huihui Zhang, Chang Xu. ✉e-mail: xiaowen.zhan@ahu.edu.cn; qluo@ustc.edu.cn; shangao@ahu.edu.cn; yxie@ustc.edu.cn

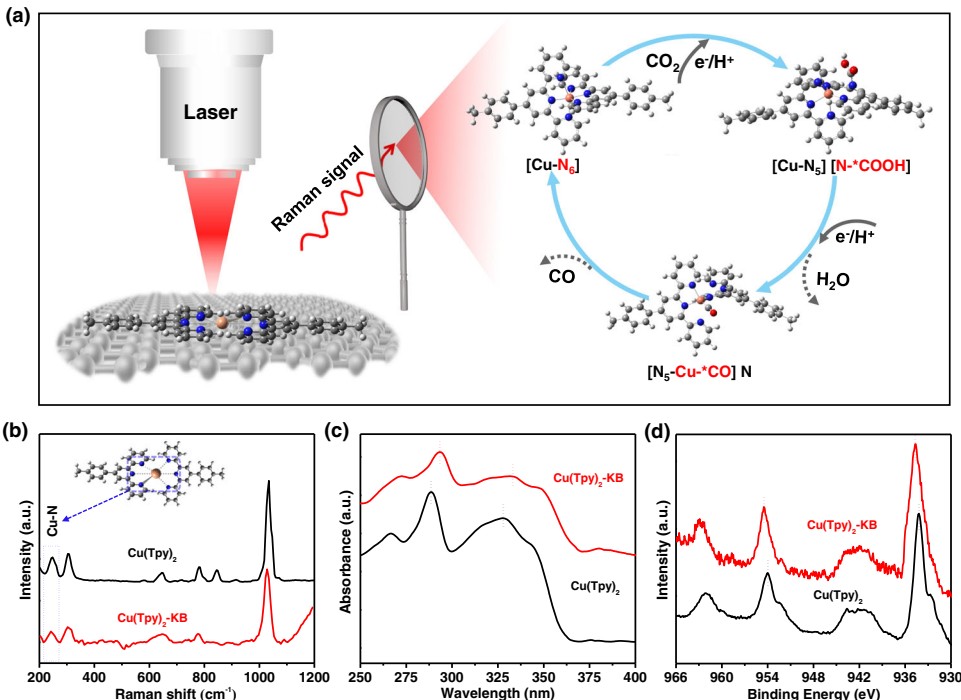

**Fig. 1 | Schematic representation of the operando Raman analysis used in this study and the physicochemical characteristics of Cu(Tpy)2 and Cu(Tpy)2-KB.** **a** A scheme illustrating the operando Raman study on the CO₂RR process of Cu(Tpy)₂-KB@GC electrocatalyst. **b** Raman spectroscopy, **c** UV-vis and (**d**) XPS Cu2*p* spectra of Cu(Tpy)₂ and Cu(Tpy)₂-KB. H atom in white, C atom in gray, N atom in blue, O atom in red and Cu atom in brown, respectively.

spectroscopy analysis has been mainly dedicated to capturing the formation and evolution of intermediates, while the dynamic behaviors of both the geometric structure and electronic environment of the catalysts, as well as their effects on CO₂RR process are largely unknown[11,12]. Until very recently, Wright *et al.* performed a thorough mechanistic study on an immobilized [Ni(2,2′:6′,2″-terpyridine-4′-thiol)₂](BF₄)₂, or Ni(tpyS)₂, complex for CO₂RR; nonetheless, their observations relied on a complex gap-plasmon-assisted SERS technique[13]. Therefore, in pursuit of wide applicability and simplicity, we are intrigued to employ a more widely accessible *operando* Raman spectroscopy (relative to SERS) in unravelling the underlying CO₂RR mechanisms, including the identification of intermediates and active sites, and more importantly, the monitoring of catalyst structure dynamics during the electrocatalytic process.

In terms of model catalyst selection, metal-polypyridyl complexes have attracted significant attention, mainly because of their high activity rooted in atomically dispersed catalytic sites and versatile coordination structures available for molecular-level engineering[14]. Apart from the early research in scarce and pricy Re- or Ru-based polypyridyl complexes[15–20], considerable recent research efforts have been extended to complexes based on more abundant and cheaper first-row transition metals, such as Mn[21,22], Co[23], and Ni[24,25]. However, simultaneously achieving high selectivity and long-term stability is still facing significant challenges, owing to insufficient mechanism understanding[14,26]. For example, the occurrence of CO₂ adsorption was believed to necessitate Ni-N bond breaking for the Ni(tpyS)₂ catalyst[13], while for the photocatalytic CO₂RR on [Ni(tpy)₂]²⁺ the detachment of one tpy ligand from Ni was suggested to be a must[25]. Such ambiguity in catalyst structure change and CO₂RR pathways for transition metal-polypyridyl hybrids calls for further studies conducted in a rigorous manner. To this end, we consider a less studied bis-tpy {[Cu(Tpy)₂]Cl₂•*x*H₂O} complexes (where tpy = 4-(4-methylphenyl)−2,2:6,2-terpyridine) in this category suitable as a model catalyst (hereinafter referred to as Cu(Tpy)₂) for this purpose of study, since it is simple, cheap and capable of flexible structure tuning for performance enhancement.

Both abovementioned initiatives considered, we herein, as illustrated in Fig. 1a, demonstrate the elucidation of CO₂RR mechanisms via a widely accessible *operando* Raman/FT-IR spectroscopy analysis coupled with density functional theory (DFT) computations at B3LYP/6-31 G** level, with the emphasis on both the reaction pathways and real-time molecular structure evolutions. As a proof of concept, we synthesize Cu(Tpy)₂ complex and immobilize it on Ketjen black (KB) (hereinafter referred to as Cu(Tpy)₂-KB) to form a model electrocatalyst capable of kilogram-scale production. After casting the Cu(Tpy)₂-KB on the glassy carbon (GC) electrode, the resulting Cu(Tpy)₂-KB@GC exhibits a high CO selectivity of 99.5%, 80 h stability at −0.6 V vs. the reversible hydrogen electrode (vs. RHE), along with exceptionally high turn-over efficiencies (TOFs), all of which are among the best values reported to date. Combining *operando* experimental and simulative spectroscopic knowledge and insights from theoretical computations, we for the first time revealed a baton-relay-like mechanism of actives sites transfer during electrocatalytic CO₂RR. Specifically, the first protonation step involved breakage of a Cu-N bond before the formation of *COOH immediate at the N site, the second protonation step of *COOH occurred at the central N active site, prior to an interesting N-to-Cu active site transfer for the subsequent *CO formation on Cu sites. This unique active site transfer allows a significantly lower energy for *CO formation on Cu site (0.18 eV) than that on central N site (1.27 eV), and enables subsequently a low-barrier CO desorption (−0.13 eV) and catalyst recovery, elucidating the high CO selectivity and catalytic durability of Cu(Tpy)₂-KB@GC. As highlighted in Fig. 1a, this work disclosed the full mechanistic picture concerning the CO₂RR process of Cu(Tpy)₂-KB@GC system, essentially based on clear evidences acquired from a widely available *operando* Raman. The method and results presented here are expected to accelerate mechanistic study in this field, leading to development of next-generation electrocatalysts through rational design.

## Results and discussion

### Preparation and characterization of Cu(Tpy)$_2$-KB

The Cu(Tpy)$_2$-KB electrocatalyst used in this study was produced in large quantity via a scalable approach. Supplementary Fig. S1 shows the detailed synthesis route and kilogram-scale demonstration of Cu(Tpy)$_2$ and Cu(Tpy)$_2$-KB). First, Cu(Tpy)$_2$ molecule was synthesized through a facile two-step method from previous reports[27,28] with modifications. In the resulting Cu(Tpy)$_2$ compound, Cu$^{2+}$ is bonded to six N atoms in a strongly distorted octahedral coordination[28]. Subsequently, a 3D carbon network formed by low cost KB is used to non-covalently immobilize Cu(Tpy)$_2$ (Supplementary Fig. S1a,b) thereby potentially offering well-defined catalytic sites and sufficient electronic conduction, a strategy proved effective for CNT-supported cobalt and nickel phthalocyanines[29,30]. The atomic-resolution high-angle annular dark-field scanning transmission electron microscopy (HAADF-STEM) image of Cu(Tpy)$_2$-KB (Supplementary Fig. S2a,b) exhibited isolated bright spots of ~0.1 nm (red circles) on the carbon matrix, indicative of the presence of isolated Cu atoms. Energy-dispersive X-ray spectroscopy elemental mappings of Cu, C and N confirmed the uniform molecular-level dispersion of Cu(Tpy)$_2$ in the carbon network (Supplementary Fig. S2c–f).

Raman spectroscopy, ultraviolet-visible spectroscopy (UV-vis), and X-ray photoelectron spectroscopy (XPS) were further conducted to shed lights into potential molecular-carbon interactions in the Cu(Tpy)$_2$-KB hybrid. As shown in Fig. 1b, the Raman spectrum of Cu(Tpy)$_2$ shows some vibrational features, i.e., at 780 and 844 cm$^{-1}$, that are missing or considerably depressed in that of Cu(Tpy)$_2$-KB, suggesting that electronic interactions between Cu(Tpy)$_2$ and KB inhibited some vibration modes of Cu(Tpy)$_2$[31]. Additional evidence can be found in the UV-vis spectra of Cu(Tpy)$_2$ and Cu(Tpy)$_2$-KB displayed in Fig. 1c. The two absorbance bands near 268 nm and 330 nm shared by the two materials correspond to the metal-to-ligand charge transition and d-d transition, respectively[32]. However, clear red shifts of both bands occur in the spectrum of Cu(Tpy)$_2$-KB, which can serve as an indicator of the electronic coupling between Cu(Tpy)$_2$ and KB. The idea that such interaction, i.e., π-π stacking[29,31,33–35], occurred during the immobilization process is reinforced by the XPS results shown in Fig. 1d. Specifically, the two Cu2$p$ characteristic peaks of Cu(Tpy)$_2$-KB shift to higher energies relative to those of Cu(Tpy)$_2$, strongly pointing to a bonding environment change for Cu$^{2+}$. Therefore, all the aforementioned results strongly confirmed the successful synthesis of Cu(Tpy)$_2$-KB electrocatalyst in large quantity via a scalable approach.

We next extend the benefits of anchoring molecules on carbon framework by including surface wettability as an important aspect for CO$_2$RR. The water drop attached on the catalyst surface layer formed a contact angle of 32.1° for Cu(Tpy)$_2$-KB and 76.4° for Cu(Tpy)$_2$ (Supplementary Fig. S3), suggesting the former exhibited higher hydrophilicity. The hydrophilic Cu(Tpy)$_2$-KB electrode can afford a larger electrochemical active area and a lower adhesion force to a gas bubble (i.e., of CO) when compared with Cu(Tpy)$_2$. This not only allows fast charge transfer reaction occurring at small overpotentials and large currents, but also leads to a quick release of CO gas bubbles from the electrode surface to ensure intimate catalyst/electrolyte contact[36,37]. Supplementary Fig. S4 shows the CO$_2$ adsorption isotherms of neat KB, Cu(Tpy)$_2$, and Cu(Tpy)$_2$-KB measured at 25 °C. Cu(Tpy)$_2$-KB demonstrates the highest CO$_2$ uptake among the three, which is beneficial for achieving good CO$_2$RR performance. As a result, the promising hydrophilicity and CO$_2$ adsorption of Cu(Tpy)$_2$-KB electrocatalyst could continually provide raw materials to trigger CO$_2$ reduction into energy-rich fuels.

### Electrocatalytic performance of Cu(Tpy)$_2$-KB@GC

Before exploring Cu(Tpy)$_2$-KB@GC's catalytic activity, we tested the catalytic performance of the Cu(Tpy)$_2$@GC under CO$_2$ in DMF/H$_2$O (95:5, v-v) with 0.1 M of TBAP as supporting electrolyte. As shown in Supplementary Fig. S5, Cu(Tpy)$_2$@GC afforded a notable cathodic current response and a good CO$_2$-to-CO selectivity with a high FE(CO) of 95.5% at −0.6 V vs. RHE. However, during the chronoamperometry test performed at −0.6 V vs. RHE in the organic electrolyte system, both the current density and FE(CO) gradually decreased, suggesting the deactivation of Cu(Tpy)$_2$@GC (Supplementary Fig. S6). It is known that the cation in the electrolyte plays a role in the activity and selectivity of electrocatalysts[38–48]. Linear-sweep voltammetry (LSV) of Cu(Tpy)$_2$-KB@GC was first performed by using CO$_2$-saturated 0.5 M aqueous solutions of various alkali bicarbonates (Li, Na, K). The trend for the current density is Li$^+$ < Na$^+$ < K$^+$, while their faradic efficiency values (FEs) for CO production are very close (Supplementary Fig. S7a, b). The higher catalytic currents in the case of KHCO$_3$ is likely due to the faster migration of K$^+$ ions than Na$^+$ and Li$^+$ in aqueous solutions, as K$^+$ ions have a weaker hydration/solvation effect[40]. Based on this observation, 0.5 M KHCO$_3$ was used as the electrolyte in this study. Supplementary Fig. S7c,d compared the electrocatalytic performances of Cu(Tpy)$_2$ catalysts immobilized on four different carbon supports. Clearly, the carbon support type exerts little influence on the catalytic activity and CO selectivity. Therefore, we selected Cu(Tpy)$_2$ with the KB support for further study considering its slightly higher FE(CO) output. The LSV results for Cu(Tpy)$_2$@GC and Cu(Tpy)$_2$-KB@GC were compared in Fig. 2a. The CO$_2$ reduction products were characterized by gas chromatography and differential electrochemical mass spectrometry (DEMS) measurements at various applied potentials (see Supplementary Fig. S8 and Fig. S9 for data at −0.4 V vs. RHE as examples). Only two gas-phase products (i.e., CO and H$_2$) were identified, and no liquid phase products were detected. In a CO$_2$-saturated 0.5 M KHCO$_3$ electrolyte, CO could be detected at a very small onset overpotential of 0.29 V (−0.4 V vs. RHE) for Cu(Tpy)$_2$-KB@GC. Cu(Tpy)$_2$-KB@GC showed very attractive electrocatalytic activity, with its final current density reaching as high as ~53 mA/cm$^2$ at −1.0 V vs. RHE. The FEs for CO production at a fixed potential of −0.6 V vs. RHE of different materials were compared in Fig. 2b. Expectedly, both Tpy@GC and KB@GC witnessed no CO production. When the copper active sites were incorporated, the resulting Cu(Tpy)$_2$@GC started to demonstrate electrocatalytic activity toward CO$_2$-to-CO with a noticeable FE(CO) of 9.6% at −0.6 V vs. RHE. This suggests that Cu$^{2+}$ and its strongly distorted octahedral coordination with terpyridine ligand can effectively catalyze CO production. Encouragingly, successful carbon immobilization further enhanced the catalytic activity, unleashing a high FE(CO) of 99.5% for Cu(Tpy)$_2$-KB@GC at −0.6 V vs. RHE. As shown in Fig. 2c, the FE(CO) for Cu(Tpy)$_2$-KB@GC is higher than that for Cu(Tpy)$_2$@GC at each tested potential, suggesting high CO selectivity of Cu(Tpy)$_2$-KB@GC. TOFs for CO production were calculated for Cu(Tpy)$_2$-KB@GC as shown in Fig. 2d (see Methods section for calculation details). Notably, TOFs(CO) of Cu(Tpy)$_2$-KB@GC reached as high as 9.4 s$^{-1}$ at −0.6 V vs. RHE and 38 s$^{-1}$ at −0.8 V vs. RHE, both of which are among the best values reported in the context of molecular electrocatalysts (see Table S1 for a detailed comparison with literature data).

To provide kinetic evidences for the CO$_2$RR catalytic activity of Cu(Tpy)$_2$-KB@GC, we measured the partial current density for Tafel plots of Cu(Tpy)$_2$@GC and Cu(Tpy)$_2$-KB@GC catalysts (Supplementary Fig. S10). The fact that Cu(Tpy)$_2$@GC exhibited a much higher value of 254 mV dec$^{-1}$ implies that Cu(Tpy)$_2$-KB@GC exhibited much faster reaction kinetics than Cu(Tpy)$_2$@GC[49,50]. This assertion was strengthened by the electrochemical impedance spectra results shown in Supplementary Fig. S11. Evidently, Cu(Tpy)$_2$-KB@GC shows a considerably lower charge transfer resistance than Cu(Tpy)$_2$@GC, indicative of lower kinetic barriers for the catalytic reaction occurring on the electrode surface. Overall, the faster electrocatalytic kinetics of Cu(Tpy)$_2$-KB@GC can be favorably linked to (i) the well-defined catalytic sites (free of micrometer-sized aggregates), (ii) the considerably

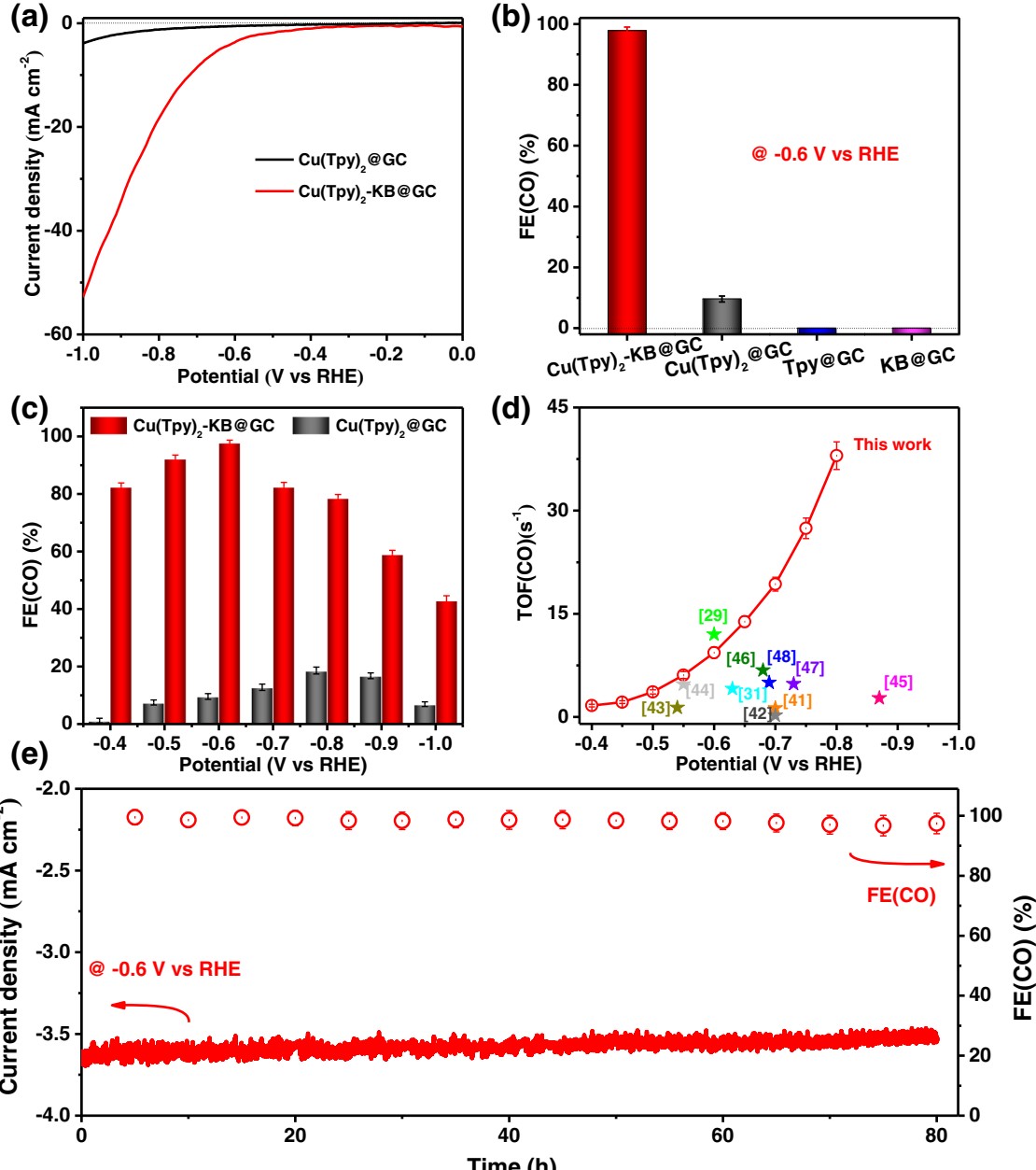

**Fig. 2 | CO2RR performance of Cu(Tpy)2@GC and Cu(Tpy)2-KB@GC electrocatalysts. a** Linear sweep voltammetry of neat Cu(Tpy)$_2$@GC and Cu(Tpy)$_2$-KB@GC in a CO$_2$-saturated 0.5 M KHCO$_3$ electrolyte. **b** FEs for CO and H$_2$ production of neat KB@GC, Tpy@GC, Cu(Tpy)$_2$@GC and Cu(Tpy)$_2$-KB@GC. **c** FE(CO)s of Cu(Tpy)$_2$@GC and Cu(Tpy)$_2$-KB@GC at different applied potentials. **d** TOFs calculated for Cu(Tpy)$_2$-KB@GC at different potentials. **e** Chronoamperometry and FEs for CO formation at a fixed potential of −0.6 V vs. RHE for Cu(Tpy)$_2$-KB@GC. Error bars represent the standard deviations from three independent measurements.

enhanced electronic conductivity, and (iii) the sufficient electrochemical active area and low adhesion force to a gas bubble (i.e., of CO) benefiting from the improved catalyst/electrolyte contact (Supplementary Fig. S4).

Chronoamperometry was further conducted to evaluate the long-term stability. At a fixed potential of −0.6 V vs. RHE, Cu(Tpy)$_2$-KB@GC sustained a stable operation for 80 h with a high initial FE$_{(CO)}$ of 99.5% and >96% FEs$_{(CO)}$ throughout the whole process (Fig. 2e). As shown in Supplementary Fig. S12, the CO production of Cu(Tpy)$_2$-KB@GC ended up at 375 mmol. The impressive stability of Cu(Tpy)$_2$-KB@GC was corroborated by our post-electrolysis analysis. As shown in Supplementary Fig. S13a, the major FT-IR peaks of Cu(Tpy)$_2$-KB@GC before and after electrolysis remain unchanged, indicating no obvious structural change; meanwhile, the Cu2$p$ XPS spectra of Cu(Tpy)$_2$-

KB@GC are also identical before and after the chronoamperometry test, with no sign of valence state change for Cu over the course of electrocatalysis (Supplementary Fig. S13b). HAADF-STEM images of Cu(Tpy)$_2$-KB@GC before and after electrolysis were compared in Supplementary Fig. S14, disclosing that the uniform molecular-level dispersion of Cu(Tpy)$_2$ in the carbon network was well maintained, with no sign of Cu sites converting into metallic copper agglomerates. It is worth noting the reversible structural change of some Cu-based molecular catalysts has been confirmed by operando techniques, namely X-ray absorption spectroscopy[51–53]. For instance, Karapinar[51] and Wang[52] et al. reported that the isolated Cu sites in their copper-complex catalysts would, during CO$_2$RR, transiently convert into metallic copper nanoparticles that are likely to be the catalytically active species. This process was reversible and the initial material

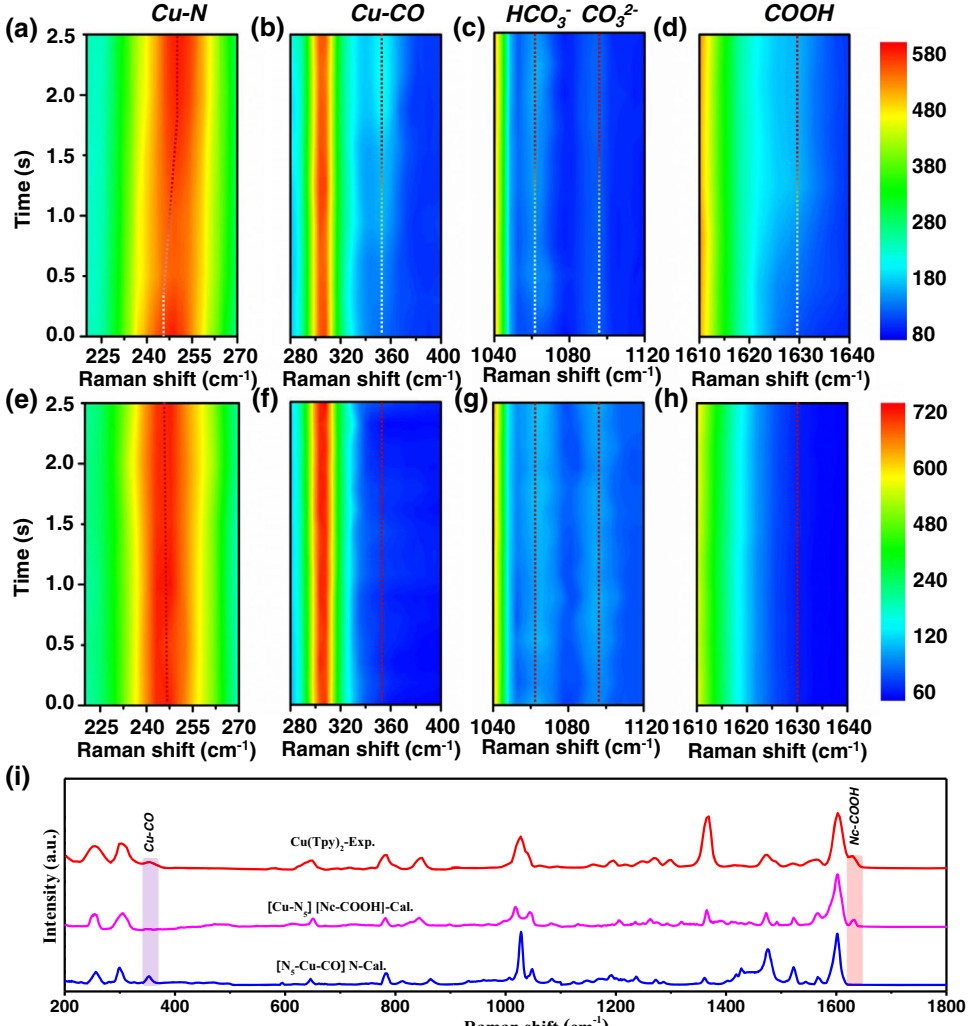

**Fig. 3 | Time-dependent operando Raman spectra of Cu(Tpy)2-KB@GC under electrocatalytic conditions.** The spectra were measured using (**a**–**d**) a $CO_2$-saturated 0.5 M $KHCO_3$ solution and (**e**–**h**) an Ar-saturated 0.5 M $KHCO_3$ solution, respectively. The potential was controlled at −0.45 V vs. RHE. The results are displayed in three scanning subsections, namely (**a**, **e**) 220-270 $cm^{-1}$, **b**, **f** 280–400 $cm^{-1}$, **c**, **g** 1040–1120 $cm^{-1}$, and (**d**, **h**) 1610–1640 $cm^{-1}$. **i** Comparisons among the DFT-calculated and *operando* Raman spectra of Cu(Tpy)2. The experimental line was extracted from the top spectra in (**a**–**d**) at the end of 2.5 s.

could be recovered intact after electrolysis. However, not all copper complexes restructure reversibly during $CO_2RR$. In other studies, the Cu atoms in molecular complexes were reduced from Cu(II) to metallic Cu(0) which later agglomerated into clusters, indicating an irreversible restructuring[54,55]. We further explored the stability of Cu(Tpy)2-KB@GC catalyst during $CO_2RR$ at a higher potential of −1.5 V vs. RHE. The results indicate that when the potential was too high, i.e., −1.5 V vs. RHE, major conversion of $Cu^{2+}$ sites to metallic Cu nanoclusters was evident (see Supplementary Fig. S15 and Fig. S16 for details). It is now safe to conclude, based on all evidence discussed above, that under −0.6 V vs. RHE used for our long-term stability demonstration (Fig. 2e), the Cu(Tpy)2-KB@GC catalyst exbibits a high stability without irreversible reduction of Cu sites over the course of electrocatalysis.

To sum up, the above results indicate that Cu(Tpy)2-KB@GC exhibits not only high catalytic activity, but also stability up to 80 h and product selectivity ($FE_{CO}$ = 99.5%) at −0.6 vs. RHE. Supplementary Table S1 presented a comprehensive comparison in electrocatalytic performance of some representative molecular catalysts for $CO_2$-to-CO conversion. Clearly, our Cu(Tpy)2-KB@GC is among the top few noble-metal-free $CO_2$-to-CO catalysts reported so far to simultaneously afford a fairly high current density, high TOFs(CO), and a durability up to 80 h.

## *Operando* spectrochemistry evidence for active sites transfer

To monitor the evolution of catalyst structure and surface adsorbates of Cu(Tpy)2-KB@GC during $CO_2RR$, *operando* Raman spectroscopy analysis was conducted. The Raman spectra were collected in every 0.1 s from 0 to 2.5 s under a static scanning mode at a controlled potential of −0.45 V vs. RHE. As shown in Fig. 3a–d, the peaks at 247, 354, and 1630 $cm^{-1}$ can be assigned to the Cu-N bonding environment in the hybrid, *CO, and *COOH, respectively. The assignment of Cu-N peak was in good agreement with the data shown in Supplementary Fig. S17, where the Raman lines of the pristine Tpy and Cu(Tpy)2 are compared.

For the convenience of discussion, we divide the data into three stages based on the general trend. At the first stage (0–0.4 s) of detection, no *CO or *COOH components were detected, and there existed little change for the Cu-N peak position. Moving to the second stage (0.4–2.2 s), *COOH and *CO species appeared and their respective peaks gradually intensified. This can serve as an evidence that the $CO_2$ was absorbed and subsequently electrocatalytically reduced on the surface of Cu(Tpy)2 complex, with *COOH and *CO as the intermediates. Further increasing the electrolysis time witnessed no obvious change during the third stage (2.2–2.5 s), recommending that the catalytic equilibrium was reached. As a confirmation for the

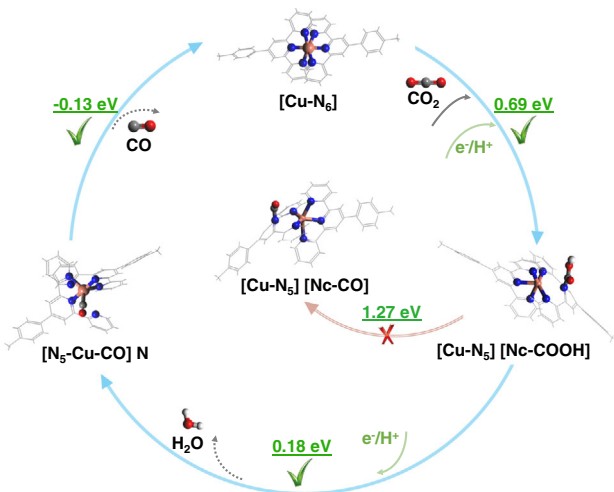

**Fig. 4 | A baton-relay-like mechanism of actives sites transfer for the electro-catalytic CO2RR to CO on Cu(Tpy)2-KB@GC catalyst.** H atoms are in white, C atoms in gray, N atoms in blue, O atoms in red and Cu atoms in brown, respectively. Free energy change (ΔG) values for key bifurcating intermediates are annotated in eV.

occurrence of CO₂RR reaction, similar *operando* measurements were conducted in a "blank" electrolyte, namely, Ar-saturated 0.5 M KHCO₃ solution. As can be seen in Fig. 3e–h, over the course of measurements all Raman peaks remained unchanged, indicating the absence of CO₂RR process in Ar-saturated 0.5 M KHCO₃ solution, and more importantly, that the peak evolution of Cu-N, *CO, *COOH observed in Fig. 3a–d unambiguously originated from the CO₂ reduction process.

One important takeaway from Fig. 3a–d is that the Cu-N peak clearly shifted to the right at the second stage, during which the augment of *COOH and *CO intermediates peaks occurred. We favorably link this continuous Cu-N peak shift to the (i) chemical adsorption of CO₂ to the N site that is coordinated to the center Cu atom of Cu(Tpy)₂-KB@GC, and (ii) subsequent breaking of Cu-N bonds. Such Cu-N Raman peak shift resulting from bonding environment changes, i.e., the introduction of function groups[56] and temperature-induced structure changes[57], have been widely reported in the literature. More importantly, the N site also offers a suitable protonation site for the adsorbed CO₂ molecule thereby facilitating the formation of *COOH intermediate[3], which aligns well with the rise of *COOH peak in Fig. 3d. Therefore, we speculated the CO₂ adsorption and subsequent *COOH formation occurred at the N active site. The second important observation to follow is that during the second stage *CO adsorbed on the Cu active site rose as another intermediate specie (Fig. 3a). This evidence pointed to a possible transfer of the active center from the N to Cu site, where the *COOH was attacked by a proton-electron to release one H₂O molecule and form *CO.

To provide additional evidence, *Operando* FT-IR spectra of Cu(Tpy)₂-KB@GC under electrocatalytic conditions were acquired using a similar protocol as used for Raman analysis. The spectra were collected respectively in a CO₂-saturated 0.5 M KHCO₃ solution (Supplementary Fig. S18a–c) and an Ar-saturated 0.5 M KHCO₃ solution (Supplementary Fig. S18d–f). We observed a clear peak evolving gradually at -1424 cm⁻¹ only in the presence of CO₂-saturated electrolyte, which unarguably pointed to the *COOH intermediate formation[58] and further corroborated the Raman observation of *COOH in Fig. 3a.

To clarify the two key experimental Raman spectral features, we next performed DFT simulations of Cu(Tpy)₂ in the corresponding vibrational frequency range (see Methods section for Computational details). To demonstrate the reliability of our simulation, the calculated spectrum was compared in line with the experimental line of the pristine Cu(Tpy)₂ (Supplementary Fig. S19). The detailed peak

assignments for experimental and simulated Raman spectra can be found in Supplementary Tables S2–S4. Overall, the two spectra show good consistency in terms of major peak positions in spite of some peak displacements mostly likely rising form the unavoidable discrepancies between the structures used for modeling and for actual measurements. Two simulated Raman lines are shown Fig. 3i, namely, [Cu-N₅][Nc-COOH]-Cal. and [N₅-Cu-CO]N-Cal., corresponding to two states where the *COOH intermediate was formed on the central N site (Nc) and the *CO intermediate was formed on the Cu site, respectively. The experimental spectrum (Exp.) extracted from the 0.25 s of Fig. 3a–d was displayed here for comparison. We should note here that the experimental data contained all vibrational features existed on the catalyst surface, while the simulated data characterized two special states. Bearing this in mind, good agreements were met among the simulated and experimental data. Specifically, [Cu-N₅][Nc-COOH]-Cal. and Exp. display Nc-COOH features at the same position, while [N₅-Cu-CO] N-Cal. and Exp. share the Cu-CO peak. This accordance was further strengthened by including simulations performed for two alternative situations into consideration. The two cases are (i) the *COOH intermediate was adsorbed on either of the two side N sites, namely [Cu-N₅][N₅₁-COOH]-Cal. or [Cu-N₅][N₅₂-COOH]-Cal., and (ii) the *CO intermediate was formed on the central N site, namely [Cu-N₅][N₅-CO]-Cal., without N-to-Cu active center change. The geometric positions, Mulliken charges of *COOH and Gibbs free energies (ΔG) calculated for the *COOH formation reaction are displayed in Supplementary Fig. S20 and Supplementary Table S5 for the three different N sites. As shown in Supplementary Fig. S21, neither of the two situations exhibited peaks matching the Nc-COOH peak from the experimental line, which successfully ruled out these possibilities. It is worth mentioning that we carefully checked all possible adsorption patterns of COOH species, especially the bridge-like model[59] as Cu-OOHC-N; however, it could not be formed in this system most likely owing to the large ring strain.

### Proposed mechanistic pathways of CO₂ reduction to CO

The corresponding full reaction mechanism is demonstrated in Fig. 4 with the calculated Gibbs free energy labeled for each potential pathway. Overall, the results indicate a two-step proton-electron (H⁺/e⁻) transfer process obeying the baton-relay-like N-to-Cu site-transfer mechanism. The stepwise discussion on the proposed reaction mechanism is presented below.

In the first protonation step to form the carboxyl intermediate *COOH, the two possible competitive reactions are the hydrogen evolution reaction (HER) to form H₂ and the formation of the formate *HCOO species (if the proton attacks the O in CO₂). However, the computational results (Supplementary Fig. S22) clearly reveal that both ways are energetically less favorable, with 2.44 eV, 0.85 eV and 0.69 eV for the *HCOO, *H and *COOH step, respectively. This was also confirmed by both the operando and simulated Raman spectra (Fig. 3d, i). As revealed by the Raman analysis, it is interesting to note that *COOH is adsorbed on an active N center rather than the Cu site, in which both electronic and geometric characteristics play important roles (see Supplementary Table S5). For one thing, the central N site (Nc) with a negative charge state (−0.73 |e| of Mulliken population) is deduced to be the suitable protonated site, other than the Cu site with a positive charge (+0.86 |e| of Mulliken population). Moreover, among the three different N sites in Cu(Tpy)₂, adsorption on the central N atom with the most negative charge state and lowest Gibbs free energy change (0.69 eV) is absolutely more favorable in the COOH* formation step (see Supplementary Fig. S20 and Table S5). For another thing, the existence of the steric hindrance effect between *COOH and the catalyst also makes the adsorption on N sites more feasible than the Cu site. It is also worth mentioning that the coordination number of Cu is reduced to five-fold along with the adsorption of *COOH, as the N-COOH covalence

interaction is formed which localizes the valence electrons in this N atom and breaks the Cu-N coordination bond. The adsorption of *COOH and breaking of the Cu-N bond are responsible for the continuous Cu-N Raman peak shift as revealed in Fig. 3a.

For the second protonation step, the *COOH stays attached on the Nc site without N-to-Cu site transfer, the free energy will rise dramatically to 1.27 eV, which is much higher than that of the case involving the site transfer (0.18 eV). Following this favorable route, *COOH is attacked by another proton-electron to proceed with the second protonation step, with one $H_2O$ molecule released and *CO generated at the Cu site. Obviously, *CO adsorption species is formed on the Cu site in this energetic favorable route, accompanied with the transfer of active site, as CO is a suitable ligand for Cu, and sixfold coordinated Cu is regained in this process. This result was also confirmed by the experimental Raman spectra as shown in Fig. 3i.

Finally, the *CO intermediate is dissociated from $Cu(Tpy)_2$ and catalytic agent is regenerated. The coordination pattern undergoes the transformation from [Cu-$N_6$] (catalytic agent) to [Cu-$N_5$] (N-COOH), to [$N_5$-Cu-CO] N during the reaction process, as summarized in Fig. 4. It should be noted that, a small energy barrier of 0.48 eV was required for the *CO desorption process where a transition state exists (see Supplementary Fig. S23). We would like to emphasize that the N-to-Cu site transfer is vital for this favorable $CO_2$RR-to-CO route to realize high CO selectivity, in that it offers a huge energy reduction of 1.09 eV for *CO formation (1.27 eV on N vs. 0.18 eV on Cu). In short, the proposed baton-relay-like Nc-to-Cu site-transfer reaction pathway, deduced from rigorous operando spectroscopic and computational analysis, provides a significant explanation why $Cu(Tpy)_2$-KB@GC affords high catalytic activity and CO selectivity in the electrochemical $CO_2$RR.

In summary, a carbon-supported copper-terpyridine model electrocatalyst capable of kilogram-scale production was introduced as a cost-effective electrocatalytic $CO_2$RR catalyst. It enabled a high FE(CO) of 99.5%, an 80 h stability, and a turn-over efficiency of 9.4 s$^{-1}$ at −0.6 V vs. RHE. For the first time, a commonly available *operando* Raman/FT-IR spectroscopy analysis assisted with DFT calculations was used to directly observe the active sites and elucidate the conversion pathways for $CO_2$RR. Remarkably, a baton-relay-like site-transfer mechanism was revealed by operando spectrochemistry analysis. Specifically, the $CO_2$ adsorption involved breakage of a Cu-N bond before the formation of *COOH immediate exclusively at the central N active site, which was followed by an interesting N-to-Cu active site transfer for the subsequent *CO formation on Cu. Significantly, the unique site transfer is key to achieving highly selective and stable CO production, in that it enables an energetically favorable *CO formation on Cu site (0.18 eV, in contrast to 1.27 eV on N site), and subsequently a low-barrier CO desorption process (−0.13 eV) and a catalyst recovery without irreversible structure and morphology changes of Cu sites.

We believe the present work is of high value to future research, essentially in that we successfully demonstrate the capability of a generally accessible operando spectroscopy in unraveling evolution of both geometric structure and electronic environment of the catalytic sites thereby elucidating complex $CO_2$RR mechanisms indispensible for tuning electrocatalytic performance. The approach and findings here can not only provide in-depth insights in rationally designing efficient transition-metal complex catalysts for electrochemical $CO_2$ reduction, but more importantly, establish a new platform for worldwide extensive mechanistic study toward accelerating development of next-generation $CO_2$RR electrocatalysts.

## Methods

### Chemicals

4-Methylbenzaldehyde, 2-Acetylpyridine, KOH, $NH_3 \cdot H_2O$ (25%), $CuCl_2 \cdot 2H_2O$ were purchased from Sigma Aldrich and used as received.

Ketjen black (KB) was purchased from EC-600JD, Japan. The purification of KB was done by calcining the KBs at 400 °C in Ar for 3 h. Other solvents are of analytical grade and used without further purification.

### Preparation of the 4-(4-Methylphenyl)-2,2:6,2-Terpyridine (Tpy)

One-pot procedures were used for the preparation of 4-(4-Methyl-phenyl)-2,2:6,2-Terpyridine (Tpy). The reaction process can be described as below.

Specifically, 4-Methylbenzaldehyde (15 g, 125 mmol) was placed in a 500 mL round bottom flask containing 2-Acetylpyridine (45 g, 375 mmol) and 70 mL ethanol. The reaction mixture was stirred at 25 °C for 10 min. Meanwhile, a solution with potassium hydroxide (21 g, 375 mmol) dissolved in water (30 mL) was prepared and cooled to 25 °C before it was added to the above reaction mixture. Subsequently, 80 mL ammonium hydroxide (25%) was added slowly to yield a yellowish-green solid precipitation. Finally, the mixture was stirred with ethanol reflux at 75 °C for 10 h, and the purified Tpy were collected after filtration followed by washing with ultrapure water and ethanol for over three times (yield: 37 g, 92%). The basic characterization results of Tpy, including NMR, ESI-Mass and FT-IR spectra can be found in Supplementary Note 1.

### Preparation of the [Cu(Tpy)$_2$]Cl$_2 \bullet$xH$_2$O

The synthesis route can be expressed as below:

To obtain [Cu(Tpy)$_2$]Cl$_2 \bullet$xH$_2$O complex, a hot methanolic solution of Tpy (22.6 g, 70 mmol) was added slowly with constant stirring to a hot methanolic solution of CuCl$_2 \cdot$2H$_2$O (5.3 g, 35 mmol), and the resulting solution was methanol-refluxed for 2 h. The reaction mixture was cooled to 25 °C and the resulting solid was collected by filtration. The solid product was further washed with diethyl ether, vacuum dried, and recrystallized from methanol to obtain [Cu(Tpy)$_2$]Cl$_2 \bullet$xH$_2$O compound (yield: 22.4 g 90%). The basic characterization results of [Cu(Tpy)$_2$]Cl$_2 \bullet$xH$_2$O, including ESI-Mass and FT-IR spectra can be found in Supplementary Note 2.

### Preparation of the Cu(Tpy)$_2$-KB

93 g of purified Ketjen Black (KB) were dispersed in 9.3 L DMF. Then, 20 g of [Cu(Tpy)$_2$]Cl$_2 \bullet$xH$_2$O dissolved in 2 L DMF was added to KB suspension to obtain a well-mixed suspension. The mixed suspension was stirred at 25 °C for 24 h. The Cu(Tpy)$_2$-KB materials were separated by centrifuge and washed with DMF, ethanol and water, followed by lyophilization to obtain the final products.

## Preparation of the Cu(Tpy)$_2$-KB@GC, Cu(Tpy)$_2$@GC, KB@GC, Tpy@GC

Calculated amount of Cu(Tpy)$_2$-KB, Cu(Tpy)$_2$, KB, Tpy was sonicated for 2 h in 1.0 mL ethonal and 0.1 mL Nafion®117 solution, respectively. Subsequently, 3 μL catalyst ink was applied onto a GC electrode and allowed to dry in air, achieving the catalysts loading of 0.13 mg cm$^{-2}$. A CO$_2$-saturated electrolyte was prepared by passing CO$_2$ into 0.5 M KHCO$_3$ aqueous solution for 30 min.

## Physicochemical characterization

The UV-visible spectra were measured using a U-4100 photodiode array spectrophotometer. The Raman spectroscopy was performed by a Thermo Scientific Xplora Plus confocal spectrometer with an Olympus BX43 microscope. Transmission electron microscopy (TEM) and high-resolution TEM images were obtained on a JEM-2100 transmission electron microscope (JEOL). The high-angle annular dark field-scanning transmission electron microscopy (HAADF-STEM) and elemental mapping EDS were carried out by a FEI Titan Themis Z with double spherical aberration corrector, operated at 300 kV. X-ray photoelectron spectra (XPS) was measured using a ESCALAB 250Xi spectrometer with an Al anode (Al Kα = 1846.6 eV). Copper content in the catalysts were determined using ICP-AES (Thermo Scientific iCAP 6300 duo device) with H$_2$SO$_4$ solution used for the digestion of graphitic structures. Contact angle (CA) were measured using a Droplet Shape Analyzer (DSA30S, Kruss) with distilled water. Online DEMS measurements were performed using a Hiden HPR-40 mass spectrometer with a three-electrode cell configuration.

## Operando Raman measurements

The CO$_2$RR on the electrodes was performed using an *operando* Raman electrolysis cell filled with CO$_2$- or Ar-saturated 0.5 M KHCO$_3$ electrolyte on an Olympus BX40 system with a 50× long working-length objective (HORIBA scientific, France). CO$_2$ or Ar was continuously bubbled into the electrolyte. The wavelength of the excitation laser was 532 nm.

## Operando FT-IR measurements

*Operando* FT-IR measurement was run on the catalyst covered on GC working electrode using a Nicolet iS50 Fourier-transform infrared spectrometer (Thermo Scientific, Warsaw, Poland). A Teflon cell was mounted onto a single bounce 45° Si crystal and used in a three-electrode configuration with a GC as the working electrode, a coiled Pt wire as the counter electrode and an Ag/AgCl as the reference electrode.

## Electrochemical testing

All electrochemical measurements were conducted on a CHI 760E electrochemical workstation using a three-electrode configuration with an Ag/AgCl reference electrode, a Pt foil counter electrode, and a GC disk working electrode (0.3 cm diameter, Model CHI 104, CH Instruments Inc., USA). The supporting material of GC is Kel-F. Before the electrocatalytic test, the GC was pretreated as detailed below. The surface of GC was cleaned by polishing with MicroPolish Powder 0.05 micron (CH Instruments, Inc.). Then the processed GC electrode was placed in an aqueous solution containing 1 mM K$_3$Fe(CN)$_6$ and 0.1 M KCl, and its cyclic voltammetry curve was observed. If the anodic and cathodic peaks are symmetrical with identical peak current values ($I_{PC}$/$I_{PA}$ = 1), and the peak-to-peak potential difference (Δ$E$p) was about 60 mV, then the electrode surface was considered to be well processed, otherwise it needs to be re-polished to reach the requirement. Subsequently, calculated amount of catalyst ink was deposited on the GC electrode (referred to as catalyst@GC). A CO$_2$-saturated electrolyte was prepared by passing CO$_2$ into 0.5 M KHCO$_3$ aqueous solution for 30 min. The H-type cell set-up consists of a gas-tight two-compartment

electrochemical cell separated by an anion-exchange membrane. The electrode potential was converted to reversible hydrogen electrode (RHE) scale by $E$ (vs. RHE) = $E$ (vs. Ag/AgCl) + 0.197 + 0.059 (pH). The EIS spectroscopy measurement was carried out by applying an AC voltage with 5 mV amplitude in a frequency range from 1000 Hz to 0.1 Hz at overpotential of −0.6 V (vs. RHE) in a CO$_2$-saturated 0.5 M KHCO$_3$ electrolyte.

The FEs of the gas products were calculated using the following equation:

$$FE_{gas}(\%) = \left( \frac{V_{gas}}{V_s \times V_m} \times n \times F \right) / Q \tag{1}$$

where $V_{gas}$ is the volume of the gas-phase products, $V_s$ the volume based on calibration of the GC, $V_m$ the molar volume of gas, $n$ the number of transferred electrons for gas-phase products, $F$ the Faradic constant (96485 C mol$^{-1}$) and $Q$ the total quantity of electric charge.

The TOF in s$^{-1}$ for CO production was evaluated as follows:

$$TOF(s^{-1}) = \frac{CO_{molecule/s}}{N_{atom}} = \frac{I \times t \times FE_{CO} \times M_N}{2 \times m_{catalyst} \times W_N \times F} \tag{2}$$

$$W_N = \frac{n_N \times M_N}{m_{catalyst}} = \frac{m_{catalyst} \times W_{Cu} \times M_N}{M_{Cu} \times m_{catalyst}} \times \frac{W_{Cu} \times M_N}{M_{Cu}} \tag{3}$$

where $CO_{molecule/s}$ is the number of CO molecules produced in 1 s, $N_{atom}$ the number of nitrogen sites on the catalyst, $I$ the current density of CO$_2$RR at a given potential, $t$ the reaction time (1 s), $FE_{CO}$ the faradaic efficiency of CO, $M_N$ and $M_{Cu}$ the molecular weight of nitrogen and copper, respectively, $m_{catalyst}$ the catalyst weight, and $W_N$ and $W_{Cu}$ the mass fraction of nitrogen and copper on the catalyst, respectively, $W_{Cu}$ was obtained based on ICP-AES measurements.

## Computational details

The geometry structure of [Cu(Tpy)$_2$]Cl$_2$•$x$H$_2$O molecule with two staggered Tpy ligands and a sixfold coordination Cu atom was built. Full geometry optimizations of [Cu(Tpy)$_2$]Cl$_2$•$x$H$_2$O structure and corresponding complexes studied in this paper were performed using density-functional theory (DFT) method at the B3LYP/6-31 G** level[60–63] with Grimme D3 corrections[64], based on SMD (Solvation Model Based on Density)[65] implicit solvation model. Vibrational frequencies were also calculated to confirm they are true local minima. The gas phase Gibbs free energies ($G$) of all intermediates were calculated at 298.15 K and 1 atm to determine the reaction mechanism at the same level. Raman spectra of these intermediates were also simulated and compared with experimental results to confirm our theoretical investigations.

The electrocatalytic mechanisms were studied based on Nørskov's computational hydrogen electrode model[66]. The zero voltage was defined based on potential energy ($\mu$) of components involved in reversible hydrogen electrode at all pH, $T$ and $p$ ($\mu$(H$^+$) + $\mu$(e$^+$) = 1/2$\mu$(H$_2$) at a potential of 0 V). The pathways adopted for CO$_2$ reduction to CO in this work are listed below (the asterisks represent the active sites):

$$* + CO_2 + H^+ + e^- \rightarrow {}^*COOH$$

$${}^*COOH + H^+ + e^- \rightarrow {}^*CO + H_2O(l)$$

$${}^*CO \rightarrow * + CO$$

All DFT calculations in this paper were performed using the Gaussian 16 package[67], the Raman spectrums are visualized using Multiwfn software[68].

## Data availability

All the data that support the findings of this study are available within the paper and its Supplementary Information files, or from the corresponding author on reasonable request.

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

## Acknowledgements

The authors thank Prof. Binghui Ge at Anhui University for the measurement of HADDF-STEM images. The work was supported by the National Natural Science Foundation of China (grants. U1832189 to S.G., U21A20317, 21688102 and 22103001 to J.Y.), the research start-up fund from Anhui University (S020118002/060 to S.G., S020318008/016 to Q.L., S020318008/007 to X.Z.) and the University Synergy Innovation Program of Anhui Province (GXXT–2020-001 to Q.L.). The calculations were performed at the Supercomputing Center of Anhui University, University of Science and Technology of China and National Supercomputing Center in Shanghai.

## Author contributions

Y.X., S.G., X.W.Z., and H.H.Z. conceived and designed the experiments. H.H.Z., Y.Y., and K.F.Z. performed sample synthesis, characterization and CO2 reduction measurements. C.X., Q.Q.L., and J.L.Y. carried out the first-principles calculations. All authors contributed to data analysis and writing of this paper. H.H.Z. and C.X. contributed equally to this work.

## Competing interests

The authors declare no competing interests.
