## [Peer Review File · Nature Communications]

REVIEWER COMMENTS

Reviewer #1 (Remarks to the Author):

The manuscript presents a new molecular system based on Cu that can be employed to electroreduce CO₂ to CO when supported on carbon.

The work is carefully done and presents extensive characterization techniques, importantly the authors indicate that the material can be produced in Kg scale.

There are a few issues that deserve consideration:

1. On many catalyst the cation in the electrolyte plays a role (Monteiro Nature Catal. 2021) how does this apply here?
2. The authors claim "This suggests the six-coordinated Cu²⁺ can effectively catalyze CO production. " wouldn't it be better to say Cu²⁺ and its coordination sphere in this particular case as the role of the ligand is so important?
3. The functional needs to be indicated in the main text, saying DFT is not enough to have a feeling of errors.
4. Have the authors employed the Computational Hydrogen Electrode? What is the solvation model? Is the solid represented as we have seen there are significant changes in the Tafel slope when the solid is there.
5. The charges in the main text, to what approach correspond?
6. In the SI, Figure 1 does it correspond to calculated structures?
7. Figure S8 shall be fitted, now the line is wavy.
8. Figure S12 is claimed to be showing excellent agreement, but some of the peaks are significantly displaced.
9. The authors employ "the gas phase Gibbs free energies (G) of all intermediates were calculated at 298.15 K and 1 atm to determine the reaction mechanism at the same level." Are these conditions relevant for the electrochemical system?
10. The computed structures need to be uploaded to a suitable repository.

Reviewer #2 (Remarks to the Author):

In this manuscript the authors conducted operando Raman and IR results aiming at deciphering the catalytical mechanism for an Cu-based molecular catalyst during CO₂ electrochemical reduction.

The results appear to be quite interesting.

However, two important chains are still missing between experimental data and the conclusions the authors wish to draw.

1. The data shown in Fig. 3 was collected every 3 minutes. But the TOF of the electrocatalyst is > 10. Because of this mismatch in time scale, the slow and gradual change of the Cu-N signal in ~20 min cannot directly correlate with the evolution of intermediates in a single turnover (which happens in

less than 100 ms) as shown in Fig. 4.

It can only be possibly linked with processes that take place on a similar time scale.

2. According to the mechanism given in Fig. 4, the first step requires the largest external energy input. In this case, the first step should be the rate determining step, and the following steps should be kinetically fast. Most of the sites should remain as [Cu-N6], and the formed [Cu-N5][Nc-COOH] should quickly convert into [N5-Cu-CO][N]. The change from [N5-Cu-CO][N] back to [Cu-N6] should be even faster according to the negative energy difference. Both should be short-lived intermediates (hence the chance of observation should be quite low, especially for [N5-Cu-CO][N]) This is contradictory to the observation that *COOH and Cu-CO are observed.

(It should be noted that the energy differences should not be used directly as activation energy / barrier height. They should be estimated by carefully calculating the reaction pathway and finding the energy maximum of the transition state. However the authors provided not a single transition state energy in the calculation. Hence the barrier height is estimated using the energy differences. It is of course possible that the following steps have higher activation barriers than the first step, but this is the authors' responsibility to prove.)

Taken the above considerations together, the operando Raman and IR results suggest that the catalyst undergoes a slow structure change over ca. 20 min. But the data cannot provide details about reaction mechanism as shown in Fig. 4 (at least in its current form).

Therefore, I suggest a major revision for this manuscript before publication in Nature Communications.

Reviewer #3 (Remarks to the Author):

Peer Review

Title: Mechanistic insights into CO₂ conversion chemistry of molecular electrocatalyst using accessible operando spectrochemistry

Authors: Zhang, H.; Xu, C.; Zhan, X.; Yu, Y.; Zhang, K.; Luo, Q.; Gao, S.; Yang, J.; Xie, Y.

Summary:

This research article by Xie and coworkers describes the electroreduction of CO₂ to CO with a carbon composite electrode made from Ketjan black and copper(II) bisterpyridine. The electrode exhibits promising performance in regard to current density/TOF, selectivity, catalyst lifetime, and scalability. To the best of our knowledge, this is the first example implementing Cu(Tpy)₂²⁺ to promote electroreduction of CO₂, which makes this article both exciting and questionable. The octahedrally coordinative complex, Cu(Tpy)₂²⁺ does not seem well-suited to promote catalytic transformations, yet the authors are clearly observing high catalytic activity in the experimental data presented. This is fundamentally different from other example-comparisons which have open coordination sites to bind CO₂, such as CoTPP, CoPCc, NiPCc, etc. Operando Raman and FTIR performed by the authors here suggests that the ligand is hemilabile and non-innocent leading to the baton-like mechanism. The authors' explanation is reasonable given the data and calculations at hand, but leaves the reader wondering why is Cu(Tpy)₂²⁺ not catalytically active as a homogenous species, and why is Cu(Tpy)₂²⁺ far less active on glassy carbon. The authors provide some discussion about the effects of using ketjan black, and these are some-what different reasons than the literature precedence for multiwalled carbon nanotubes. There are lingering questions such as (1) what hinders the proposed mechanism, especially in the homogenous phase (ie. in MeCN, DMF, DCM, etc.) where lability of the ligand is likely to take place, and (2) is the proposed mechanism specific to the carbon surface or does it also operate on other carbon supports? Altogether this discussion, or lack thereof, may not disqualify the article from publication, but it would certainly represent value-added for a journal with

such a broad readership. Comments below should be addressed before reconsidering publication.

Comments:

- Pg 1, column 2. "understanding" is misspelt (understaidng).
- Figure 1a. The authors depict Cu(Tpy)₂²⁺ complex, a distorted octahedral metal complex, in a highly irregular geometry which strains the Tpy ligands to be coplanar with the hexagonal graphene-like structure. Is the π-π interaction between Tpy and the carbon surface large enough to induce such a distortion? The depiction is never-the-less misleading because it is inconsistent with the proposed mechanism (Figure 1a, Figure 4) involving the octahedral configuration.
- Citation 28 is sometimes inappropriately used as a reference for the geometric parameters regarding Cu(Tpy)₂²⁺. Please consider including <https://doi.org/10.1021/ic50180a040> as the corrected citation.
- Pg 2 column 2. The authors introduce wettability experiments, however it is not clear based on the written text or Figure S3 captions what surface Cu(Tpy)₂@C is being compared to (ie. denoted in text as Cu(Tpy)₂). May the authors clarify what the surface materials are, and consider the time dependence on the contact angle measurement. This last point is particularly important in the case of porous materials which may induce changes to contact angles over time (<https://doi.org/10.1016/j.carbon.2015.01.041>). Notably, the gas isotherm data (Figure S4) does corroborate the conclusions based also on the contact angle data (Figure S3).
- Figure S5 right. There are three signals in the GC-TCD. The signal for H₂ is labeled, but there is no reference to what eluate produces the other two signals: N₂, O₂, CO, etc.?
- Pg 2-3. The authors report significant improvement in electrocatalytic activity by introducing Ketjan black as the carbon support for Cu(Tpy)₂, and this is well supported by the electrolysis data. It is rather interesting that the Tafel analysis reveals different Tafel slopes when comparing Cu(Tpy)₂ on ketjan black versus glassy carbon (Figure S7). The authors state that the much larger Tafel slope with ketjan black is caused by faster reaction kinetics, and this is likely the case. However, Tafel slopes bare consequential meaning pertaining to the mechanism. Tafel slopes ranging from 134-167 mV dec⁻¹ have literature precedence for CO₂ electroreduction (<https://doi.org/10.1021/acscatal.8b02181>). A much higher Tafel slope, as reported by the authors here, appears to be unique. Therefore, this finding warrants further discussion because there appears to be mechanistic differences by simple virtue of the carbon support.
- There is no description in the SI or elsewhere describing the conditions used to collect EIS (ie. voltage, amplitude, frequency, electrolyte, etc). Was it collected at relevant potentials for CO₂ electroreduction? Perhaps at the open circuit potential? A Bode plot alongside the Nyquist plot would also be helpful, and a full Nyquist plot of Cu(Tpy)₂ on glassy carbon showing far end behavior at high impedance could glean important information.
- Pg 4-5, Figure 3a. The formation of N-COOH intermediate species is based on a small shift in the Cu-N signal at 247 cm⁻¹, which is also rather broad. Could this shift be an artifact of identifying the peak of this broad signal? May the authors convince the reader that this assignment of the shifting peak is real? Perhaps consider summing a weighted average of the calculated signals shown in Figure 3g.
- Figure S11a shows rather irrefutably that an -COOH species is formed during electrolysis. In Table S5 the Cu-COOH species is shown to be unfavored, even though this bonding mode is often invoked for numerous metal-based catalysts. Alternatively, the bonding mode might be "Cu-OOHC-N as in <https://doi.org/10.1021/ja039577h>. Please comment on this alternative interpretation.
- Pg 6. "Preparation of the Cu(Tpy)₂." Until now the authors refer to the copper(II) complex as Cu(Tpy)₂, but actually the complex is cationic with two chlorides to balance the charge. It is perhaps ok to mention the complex in this way as an abbreviation through-out the text, but in this section of the paper please explicitly write the exact chemical formula of the compound (ie. [Cu(Tpy)₂]Cl₂ · xH₂O).
- Pg 6. Preparation of Cu(Tpy)₂@C is detailed. Cu(Tpy)₂@"glassy carbon" is missing. Please add the missing description since there is a critical comparison in the electrocatalytic section of the paper.

Responses to the Reviewers' comments and a summary of the changes made to the manuscript: NCOMMS-22-03394-T.

Many thanks to the Reviewers for having given us valuable suggestions and concerns to improve the quality of the manuscript.

We have carefully read the Reviewers' comments that help us make this manuscript better. And we acknowledge the Reviewers' positive comments that "The manuscript **presents a new molecular system** based on Cu that can be employed to electroreduce CO₂ to CO when supported on carbon. The work is **carefully done and presents extensive characterization techniques, importantly** the authors indicate that the material can be produced in Kg scale.", "**The results appear to be quite interesting.**" and "**The electrode exhibits promising performance** in regard to current density/TOF, selectivity, catalyst lifetime, and scalability".

According to the Reviewers' useful concerns and suggestions, we have supplemented many pertinent experimental data and modified the original manuscript seriously. A list of changes made and the corresponding explanations are presented as follows in a point-to-point manner.

The entire comments from Reviewer #1

*The manuscript **presents a new molecular system** based on Cu that can be employed to electroreduce CO₂ to CO when supported on carbon. **The work is carefully done and presents extensive characterization techniques, importantly** the authors indicate that the material can be produced in Kg scale. There are a few issues that deserve consideration*

1. *On many catalyst the cation in the electrolyte plays a role (Monteiro Nature Catal. 2021) how does this apply here?*

2. *The authors claim "This suggests the six-coordinated Cu²⁺ can effectively catalyze CO production. " wouldn't it be better to say Cu²⁺ and its coordination sphere in htis particular case as the role of the ligand is so important?*

3. *The functional needs to be indicated in the main text, saying DFT is not enough to have a feeling of errors.*

4. Have the authors employed the Computational Hydrogen Electrode? What is the solvation model? Is the solid represented as we have seen there are significant changes in the Tafel slope when the solid is there.

5. The charges in the main text, to what approach correspond?

6. In the SI, Figure 1 does it correspond to calculated structures?

7. Figure S8 shall be fitted, now the line is wavy.

8. Figure S12 is claimed to be showing excellent agreement, but some of the peaks are significantly displaced.

9. The authors employ "the gas phase Gibbs free energies (G) of all intermediates were calculated at 298.15 K and 1 atm to determine the reaction mechanism at the same level." Are these conditions relevant for the electrochemical system?

10. The computed structures need to be uploaded to a suitable repository.

Response: We are grateful to your great interest and very positive comments on our present work, and we also greatly appreciate your constructive suggestions guiding the revision of our manuscript. We address the comments and questions by changes to the manuscript, as detailed point-by-point below.

Point-to-point Response to Reviewer #1:

Comment 1: On many catalysts, the cation in the electrolyte plays a role (Monteiro *Nature Catal.* 2021) how does this apply here?

Response 1: Thanks for your valuable comment. As the reviewer suggested, on many catalysts, the cation in the electrolyte plays a role in catalytic reaction (*Nature Catal.* 2021, 4, 654; *J. Am. Chem. Soc.* 2022, 144, 1589-1602; *Chin. J. Catal.* 2021, 42, 1439-1444). Various theories were proposed to explain how cations at the interface affect the activity and selectivity of electrocatalysts, including modification of the local electric field, buffering of the interfacial pH, and stabilization of reaction intermediates or hydration/solvation effect. Accordingly, we explored the cation effects on the electrocatalytic CO₂RR of Cu(Tpy)₂@C by using 0.5 M XHCO₃ (X=Li, Na, K) aqueous solutions, and the results were now provided as **Fig. N1** (also copied below). As shown in **Fig. N1a**, the LSV curves revealed that the trend for the current density is Li⁺<Na⁺<

K^+ , while very similar FEs(CO) could be observed in **Fig. N1b** for various alkali bicarbonate solutions. To clarify, a few sentences were provided now on **Page 2**: “It is known that the cation in the electrolyte plays a role in the activity and selectivity of electrocatalysts.³⁸⁻⁴⁰ Linear-sweep voltammetry (LSV) was first performed by using CO_2 -saturated 0.5 M aqueous solutions of various alkali bicarbonates (Li, Na, K). The trend for the current density is $Li^+ < Na^+ < K^+$, while their faradic efficiency values (FEs) for CO production are very close (Supplementary Fig. S5a-b). The higher catalytic currents in the case of $KHCO_3$ is likely due to the faster migration of K^+ ions than Na^+ and Li^+ in aqueous solutions, as K^+ ions have a weaker hydration/solvation effect⁴⁰. Based on this observation, 0.5 M $KHCO_3$ was used as the electrolyte in this study.”

It should be noted that **Fig. N1 (a, b)** is provided as **Fig. S5 (a, b)** in the revised Supplementary Information.

Fig. N1 (a) LSV and (b) FEs(CO) acquired using various alkali bicarbonate CO_2 -saturated aqueous solutions. (c) LSV and (d) FE(CO)s at -0.6 V vs. RHE of $Cu(Tpy)_2$ catalysts immobilized on different carbon supports including Ketjen black (KB), multi wall carbon nanotubes (MWCNTs), single-wall carbon nanotubes (CNTs), and Vulcan carbon-72R (XC) using a CO_2 -saturated 0.5 M $KHCO_3$ electrolyte.

Comment 2: *The authors claim "This suggests the six-coordinated Cu²⁺ can effectively catalyze CO production. " wouldn't it be better to say Cu²⁺ and its coordination sphere in this particular case as the role of the ligand is so important?*

Response 2: We thank the reviewer for pointing out this matter. We agree with the reviewer that Cu²⁺ and its strongly distorted octahedral coordination with terpyridine ligand plays an important role. We have now rephrased on **Page 3**: *"This suggests that Cu²⁺ and its strongly distorted octahedral coordination with terpyridine ligand can effectively catalyze CO production."*

Comment 3: *The functional needs to be indicated in the main text, saying DFT is not enough to have a feeling of errors.*

Response 3: We are very grateful for reminding us. The computational details could be found in the *Computational details* section on **Page 5** of the Supplementary Information, and a brief introduction has been added in the main text (**Page 1**). "Both abovementioned initiatives considered, we herein, as illustrated in Fig. 1a, demonstrate the elucidation of CO₂RR mechanisms via a widely accessible operando Raman/FT-IR spectroscopy analysis coupled with density functional theory (DFT) computations *at B3LYP/6-31G** level*, with the emphasis on both the reaction pathways and real-time molecular structure evolutions."

Comment 4-1): *Have the authors employed the Computational Hydrogen Electrode? What is the solvation model?*

Response 4-1): We are very grateful for reminding us. The electrocatalytic mechanisms were studied based on Nørskov's Computational Hydrogen Electrode model, as mentioned in *Computational details* on **Page 5** of the Supplementary Information. SMD (Solvation Model Based on Density) implicit solvation model was employed in our computation. Relative discussions were also provided in Computational Details (**Page 5**) of the Supplementary Information (copied below).

"Full geometry optimizations of the Cu(Tpy)₂ structure and its complexes studied in this paper were performed using density-functional theory (DFT) method at the

*B3LYP/6-31G** level¹⁻⁴ with Grimme D3 corrections⁵, based on SMD (Solvation Model Based on Density)⁶ implicit solvation model.”*

“All DFT calculations in this paper were performed using Gaussian 16 package⁸, and Raman spectrum were visualized using Multiwfn software.⁹” The newly added Refs. 5, 6, 8, and 9 added can be found on **Page 33, Supplementary Information**.

Comment 4-2): *Is the solid represented as we have seen there are significant changes in the Tafel slope when the solid is there.*

Response 4-2): Thanks for your comments! The existence of the solid (Ketjan black) led to significant changes in the Tafel slope. As for Cu(Tpy)₂ without Ketjan black, the value of 254 mV/dec was ascribed to the transport limitations (*ACS Energy Lett.* 2018, 3, 1381-1386). The smaller Tafel slope of the Cu(Tpy)₂ immobilized on Ketjan black (referred to as Cu(Tpy)₂@C) was attributed to the uniform molecular-level dispersion of Cu(Tpy)₂ molecules in the electron-conducting carbon network, which alleviated the transport limitations and improved the reaction kinetics.

Comment 5: *The charges in the main text, to what approach correspond?*

Response 5: The charges in this paper corresponded to Mulliken population analysis, and the relevant description in the main text has been modified (see **Page 5**).

“For one thing, the central N site (Nc) with a negative charge state (-0.73|e| of Mulliken population) was deduced to be the suitable protonated site, other than the Cu site with a positive charge (+0.86|e| of Mulliken population).”

Comment 6: *In the SI, Figure 1 does it correspond to calculated structures?*

Response 6: Thanks! **Figure S1** was not the calculated structure, but a schematic drawing. In order to embody a distorted octahedral metal complex of Cu(Tpy)₂, the schematic illustration for the scalable synthesis and immobilization process of Cu(Tpy)₂ were redrawn (**Fig. N2**).

It should be noted that **Fig. N2** were provided as **Fig. S1a** in the revised Supplementary Information.

Fig. N2 Schematic illustration for the scalable synthesis and immobilization process of $\text{Cu}(\text{Tpy})_2$ molecules.

Comment 7: *Figure S8 shall be fitted, now the line is wavy.*

Response 7: Thanks for pointing out this issue. As the reviewer suggested, **Fig. N3** was fitted and we have added **Fig. N3** as **Fig. S10** in the revised supporting information.

Fig. N3 CO production at a fixed potential of -0.6 V vs. RHE for $\text{Cu}(\text{Tpy})_2@C$. The error bars represent standard deviation of three measurements.

Comment 8: *Figure S12 is claimed to be showing excellent agreement, but some of the peaks are significantly displaced.*

Response 8: Thanks for your valuable comment! Due to the unavoidable differences between the modeled structure and the actual material used for measurement, and the inherent errors of the DFT calculation method, some explicit features such as relative densities cannot match exactly. We have thus modified the description on **Page 5** to be

more cautions. *“To demonstrate the reliability of our simulation, the calculated spectrum was compared in line with the experimental line of the pristine Cu(Tpy)₂ (Supplementary Fig. S15). The detailed peak assignments for experimental and simulated Raman spectra can be found in Supplementary Table S2-S4. Overall, the two spectra show good consistency in terms of major peak positions in spite of some peak displacements mostly likely rising from the unavoidable discrepancies between the structures used for modeling and for actual measurements.”*

Comment 9: *The authors employ "the gas phase Gibbs free energies (G) of all intermediates were calculated at 298.15 K and 1 atm to determine the reaction mechanism at the same level." Are these conditions relevant for the electrochemical system?*

Response 9: Thanks for your valuable question. All the electrochemical measurements were conducted at room temperature and under atmospheric pressure. Therefore, the commonly used standard conditions of 298.15 K and 1 atm were adopted.

Comment 10: The computed structures need to be uploaded to a suitable repository.

Response 10: We are very grateful for the reminder. According to your suggestion, the coordinates of the computed structures were provided as **Supplementary Note 3** in the Supplementary Information (starting from **Page 24**).

The entire comments from Reviewer #2

*In this manuscript the authors conducted operando Raman and IR results aiming at deciphering the catalytical mechanism for an Cu-based molecular catalyst during CO₂ electrochemical reduction. **The results appear to be quite interesting.** However, two important chains are still missing between experimental data and the conclusions the authors wish to draw.*

1. The data shown in Fig. 3 was collected every 3 minutes. But the TOF of the electrocatalyst is > 10. Because of this mismatch in time scale, the slow and gradual change of the Cu-N signal in ~20 min cannot directly correlate with the evolution of intermediates in a single turnover (which happens in less than 100 ms) as shown in Fig. 4. It can only be possibly linked with processes that take place on a similar time scale.

*2. According to the mechanism given in Fig. 4, the first step requires the largest external energy input. In this case, the first step should be the rate determining step, and the following steps should be kinetically fast. Most of the sites should remain as [Cu-N₆], and the formed [Cu-N₅][Nc-COOH] should quickly convert into [N₅-Cu-CO][N]. The change from [N₅-Cu-CO][N] back to [Cu-N₆] should be even faster according to the negative energy difference. Both should be short-lived intermediates (hence the chance of observation should be quite low, especially for [N₅-Cu-CO][N]). This is contradictory to the observation that *COOH and Cu-CO are observed.*

(It should be noted that the energy differences should not be used directly as activation energy / barrier height. They should be estimated by carefully calculating the reaction pathway and finding the energy maximum of the transition state. However the authors provided not a single transition state energy in the calculation. Hence the barrier height is estimated using the energy differences.

It is of course possible that the following steps have higher activation barriers than the first step, but this is the authors' responsibility to prove.)

Taken the above considerations together, the operando Raman and IR results suggest that the catalyst undergoes a slow structure change over ca. 20 min. But the data cannot provide details about reaction mechanism as shown in Fig. 4 (at least in its current form).

Therefore, I suggest a major revision for this manuscript before publication in Nature Communications.

Response: We greatly appreciate your positive comments on our work and we are grateful to your constructive suggestions guiding the revision of our manuscript. We address the comments and questions by changes to the manuscript, as detailed point-by-point below.

Point-to-point Response to Reviewer #2:

Comments 1: *The data shown in Fig. 3 was collected every 3 minutes. But the TOF of the electrocatalyst is > 10. Because of this mismatch in time scale, the slow and gradual change of the Cu-N signal in ~20 min cannot directly correlate with the evolution of intermediates in a single turnover (which happens in less than 100 ms) as shown in Fig. 4. It can only be possibly linked with processes that take place on a similar time scale.*

Response 1: Thanks for your valuable comment. First, please allow us to justify our original choice of Raman settings. For Cu(Tpy)₂ immobilized on Ketjen black, the maximum scan speed for Raman measurements we explored at a controlled potential of -0.6 V vs. RHE was 0.5 s, as the signal-to-noise ratio would be worse when the scan speed increased. However, the change of the Cu-N signal under -0.6 V vs. RHE in a single turnover occurred so fast that the signal could be hardly captured. Therefore, the Raman spectra were collected in a longer interval (every 3 minutes) under a static scanning mode at a controlled potential of -0.6 V vs. RHE in the range of 0-27 minutes. Additionally, the molecular catalyst could be reconstructed or possibly reduced to metal clusters as the CO₂RR occurred according to previous reports (*Angew. Chem. Int. Ed.* 2019, 58, 15098-15103; *Nat. Commun.* 2018, 9, 415; *Nat. Commun.* 2021, 12, 2932; *Nat. Commun.* 2019, 10, 3851), which was vital for the study of catalytic mechanisms. So a longer interval was selected to ensure the successful capture of possible structure change of Cu(Tpy)₂@C.

We totally agree with the reviewer on the TOF consideration. In order to acquire Raman data in accordance with the TOF_(CO) in terms of time scale, the Raman spectra were collected under a static scanning mode at a controlled potential of -0.45 V vs. RHE (TOF_(CO) = 2.16 s⁻¹). Because of this match in time scale, the slow and gradual change

of the Cu-N signal in ~2.5 s could now directly correlate with the evolution of intermediates in a single turnover (**Fig. N4**). And the relevant description in the main text has been modified (see **Page 4**). “*To monitor the evolution of catalyst structure and surface adsorbates of Cu(Tpy)₂@C during CO₂RR, operando Raman spectroscopy analysis was conducted. The Raman spectra were collected in every 0.1 seconds from 0-2.5 s under a static scanning mode at a controlled potential of -0.45 V vs. RHE. As shown in **Fig. 3a-d**, the peaks at 247, 354, and 1630 cm⁻¹ can be assigned to the Cu-N bonding environment in the hybrid, *CO, and *COOH, respectively. The assignment of Cu-N peak was in good agreement with the data shown in Supplementary Fig. S13, where the Raman lines of the pristine Tpy and Cu(Tpy)₂ are compared.*

*For the convenience of discussion, we divide the data into three stages based on the general trend. At the first stage (0-0.4 s) of detection, no *CO or *COOH components were detected, and there existed little change for the Cu-N peak position. Moving to the second stage (0.4-2.2 s), *COOH and *CO species appeared and their respective peaks gradually intensified. This can serve as an evidence that the CO₂ was absorbed and subsequently electrocatalytically reduced on the surface of Cu(Tpy)₂ complex, with *COOH and *CO as the intermediates. Further increasing the electrolysis time witnessed no obvious change during the third stage (2.2-2.5 s), recommending that the catalytic equilibrium was reached. As a confirmation for the occurrence of CO₂RR reaction, similar operando measurements were conducted in a “blank” electrolyte, namely, Ar-saturated 0.5 M KHCO₃ solution. As can be seen in **Fig. 3e-h**, over the course of measurements all Raman peaks remained unchanged, indicating the absence of CO₂RR process in Ar-saturated 0.5 M KHCO₃ solution, and more importantly, that the peak evolution of Cu-N, *CO, *COOH observed in **Fig. 3a-d** unambiguously originated from the CO₂ reduction process.”*

It should be noted that **Fig. N4** were provided as **Fig. 3** in the revised manuscript.

Fig. N4 Time-dependent *operando* Raman spectra of Cu(Tpy)₂ under electrocatalytic conditions. The spectra were measured using (a-d) a CO₂-saturated 0.5 M KHCO₃ solution and (e-h) an Ar-saturated 0.5 M KHCO₃ solution, respectively. The potential was controlled at -0.45 V vs. RHE. The results are displayed in three scanning subsections, namely (a, e) 220-270 cm⁻¹, (b, f) 280-400 cm⁻¹, (c, g) 1040-1120 cm⁻¹, and (d, h) 1610-1640 cm⁻¹. (i) Comparisons among the DFT-calculated and *operando* Raman spectra of Cu(Tpy)₂. The experimental line was extracted from the top spectra in panels a-d at the end of 2.5 s.”

To certify the repeatability of such phenomenon observed in a single turnover, we continued to run the time-dependent *operando* Raman measurement after the 1st 2.5 s.

As showed in **Fig. N5(a-c)**, the Raman peak of Cu-N returned to the starting position (247 cm^{-1}) when a single turnover was over. It clearly shifted to the right as the CO_2RR electrocatalysis proceeded. We repeated this procedure for many times to be cautious. As showed in **Fig. N5(d-e)**, the time-dependent *operando* Raman was measured after the 9st 2.5 s, which revealed the same results.

Figure N5 Time-dependent operando Raman spectra of $\text{Cu}(\text{Tpy})_2$ under electrocatalytic conditions were acquired after respectively the 1st and 9th 2.5 s. **(a, d)** $220\text{-}270\text{ cm}^{-1}$, **(b, e)** $280\text{-}400\text{ cm}^{-1}$, and **(c, f)** $1610\text{-}1640\text{ cm}^{-1}$.

Comments 2: According to the mechanism given in Fig.4, the first step requires the largest external energy input. In this case, the first step should be the rate determining step, and the following steps should be kinetically fast. Most of the sites should remain as $[\text{Cu-N}_6]$, and the formed $[\text{Cu-N}_5][\text{Nc-COOH}]$ should quickly convert into $[\text{N}_5\text{-Cu-CO}][\text{N}]$. The change from $[\text{N}_5\text{-Cu-CO}][\text{N}]$ back to $[\text{Cu-N}_6]$ should be even faster according to the negative energy difference. Both should be short-lived intermediates

*(hence the chance of observation should be quite low, especially for [N₅-Cu-CO][N]). This is contradictory to the observation that *COOH and Cu-CO are observed.*

(It should be noted that the energy differences should not be used directly as activation energy / barrier height. They should be estimated by carefully calculating the reaction pathway and finding the energy maximum of the transition state. However, the authors provided not a single transition state energy in the calculation. Hence the barrier height is estimated using the energy differences. It is of course possible that the following steps have higher activation barriers than the first step, but this is the authors' responsibility to prove.)

Taken the above considerations together, the operando Raman and IR results suggest that the catalyst undergoes a slow structure change over ca. 20 min. But the data cannot provide details about reaction mechanism as shown in Fig. 4 (at least in its current form). Therefore, I suggest a major revision for this manuscript before publication in Nature Communications.

Response 2: Thanks for your constructive comment. According to our calculation, the first protonation step is the potential limiting step in this reaction and forms the adsorbed *COOH species. And the formed massive *COOH stay at catalysts and are ready for the next step reaction. Our Raman spectrum captured this species. Combining with the DFT calculation, this adsorbed configuration was identified as [Nc-COOH]. Following the second electron-proton step, *CO was formed. Inspired by the reviewer's insightful suggestion, we have investigated this process in detail and found that an energy barrier of 0.48 eV was required and was facile to conquer under the working condition. The reaction path containing this transition state was given in **Fig. N6**.

The barriered CO desorption allowed the adsorbed CO* to be observed experimentally. This was mainly because the dynamic process occurred in this step consists of both Cu-CO bond breakage and Cu-N bond formation. Additionally, there were many identical active sites available in Cu(Tpy)₂@C, all of which would contribute to the signals of these key intermediates, thereby enabling the successful observation of intermediates via a widely-accessible regular Raman spectroscopy.

We would like to again thank the reviewer's comments leading us to disclose such

valuable information to improve this manuscript. We therefore have added a couple of sentences on **Page 6**: “It should be noted that, a small energy barrier of 0.48 eV was required for the *CO desorption process where a transition state exists (see Supplementary Fig. S19).” Also, **Fig. N6** is now provided as Supplementary Fig. S19. Accordingly, to be accurate we have rephrased all the descriptions of this last *CO desorption process that include “spontaneous”, wherever applicable in the manuscript.

Fig. N6 Reaction path for the electrocatalytic CO₂RR to CO on Cu(Tpy)₂@C catalyst with a transition state considered for the CO desorption step.

The entire comments from Reviewer #3

*This research article by Xie and coworkers describes the electroreduction of CO₂ to CO with a carbon composite electrode made from Ketjan black and copper(II) bisterpyridine. **The electrode exhibits promising performance** in regard to current density/TOF, selectivity, catalyst lifetime, and scalability. To the best of our knowledge, this is the first example implementing Cu(Tpy)₂²⁺ to promote electroreduction of CO₂, which makes this article both exciting and questionable. The octahedrally coordinative complex, Cu(Tpy)₂²⁺ does not seem well-suited to promote catalytic transformations, yet the authors are clearly observing high catalytic activity in the experimental data presented. This is fundamentally different from other example-comparisons which have open coordination sites to bind CO₂, such as CoTPP, CoPCc, NiPCc, etc. Operando Raman and FTIR performed by the authors here suggests that the ligand is hemilabile and non-innocent leading to the baton-like mechanism. The authors' explanation is reasonable given the data and calculations at hand, but leaves the reader wondering why is Cu(Tpy)₂²⁺ not catalytically active as a homogenous species, and why is Cu(Tpy)₂²⁺ far less active on glassy carbon. The authors provide some discussion about the effects of using ketjan black, and these are some-what different reasons than the literature precedence for multiwalled carbon nanotubes. There are lingering questions such as (1) what hinders the proposed mechanism, especially in the homogenous phase (ie. in MeCN, DMF, DCM, etc.) where lability of the ligand is likely to take place, and (2) is the proposed mechanism specific to the carbon surface or does it also operate on other carbon supports? Altogether this discussion, or lack thereof, may not disqualify the article from publication, but it would certainly represent value-added for a journal with such a broad readership. Comments below should be addressed before reconsidering publication.*

Comments:

Pg 1, column 2. "understanding" is misspelt (understaidng).

Figure 1a. The authors depict Cu(Tpy)₂²⁺ complex, a distorted octahedral metal complex, in a highly irregular geometry which strains the Tpy ligands to be coplanar with the hexagonal graphene-like structure. Is the π-π interaction between Tpy and the

carbon surface large enough to induce such a distortion? The depiction is nevertheless misleading because it is inconsistent with the proposed mechanism (Figure 1a, Figure 4) involving the octahedral configuration.

Citation 28 is sometimes inappropriately used as a reference for the geometric parameters regarding $\text{Cu}(\text{Tpy})_2^{2+}$. Please consider including <https://doi.org/10.1021/ic50180a040> as the corrected citation.

Pg 2 column 2. The authors introduce wettability experiments, however it is not clear based on the written text or Figure S3 captions what surface $\text{Cu}(\text{Tpy})_2@C$ is being compared to (ie. denoted in text as $\text{Cu}(\text{Tpy})_2$). May the authors clarify what the surface materials are, and consider the time dependence on the contact angle measurement. This last point is particularly important in the case of porous materials which may induce changes to contact angles over time (<https://doi.org/10.1016/j.carbon.2015.01.041>). Notably, the gas isotherm data (Figure S4) does corroborate the conclusions based also on the contact angle data (Figure S3).

Figure S5 right. There are three signals in the GC-TCD. The signal for H_2 is labeled, but there is no reference to what eluate produces the other two signals: N_2 , O_2 , CO , etc.?

Pg 2-3. The authors report significant improvement in electrocatalytic activity by introducing Ketjan black as the carbon support for $\text{Cu}(\text{Tpy})_2$, and this is well supported by the electrolysis data. It is rather interesting that the Tafel analysis reveals different Tafel slopes when comparing $\text{Cu}(\text{Tpy})_2$ on ketjan black versus glassy carbon (Figure S7). The authors state that the much larger Tafel slope with ketjan black is caused by faster reaction kinetics, and this is likely the case. However, Tafel slopes bare consequential meaning pertaining to the mechanism. Tafel slopes ranging from 134-167 mV dec^{-1} have literature precedence for CO_2 electroreduction (<https://doi.org/10.1021/acscatal.8b02181>). A much higher Tafel slope, as reported by the authors here, appears to be unique. Therefore, this finding warrants further discussion because there appears to be mechanistic differences by simple virtue of the carbon support.

There is no description in the SI or elsewhere describing the conditions used to collect EIS (ie. voltage, amplitude, frequency, electrolyte, etc). Was it collected at

relevant potentials for CO₂ electroreduction? Perhaps at the open circuit potential? A Bode plot alongside the Nyquist plot would also be helpful, and a full Nyquist plot of Cu(Tpy)₂ on glassy carbon showing far end behavior at high impedance could glean important information.

Pg 4-5, Figure 3a. The formation of N-COOH intermediate species is based on a small shift in the Cu-N signal at 247 cm⁻¹, which is also rather broad. Could this shift be an artifact of identifying the peak of this broad signal? May the authors convince the reader that this assignment of the shifting peak is real? Perhaps consider summing a weighted average of the calculated signals shown in Figure 3g.

Figure S11a shows rather irrefutably that an -COOH species is formed during electrolysis. In Table S5 the Cu-COOH species is shown to be unfavored, even though this bonding mode is often invoked for numerous metal-based catalysts. Alternatively, the bonding mode might be “Cu-OOHC-N as in <https://doi.org/10.1021/ja039577h>. Please comment on this alternative interpretation.

Pg 6. “Preparation of the Cu(Tpy)₂.” Until now the authors refer to the copper(II) complex as Cu(Tpy)₂, but actually the complex is cationic with two chlorides to balance the charge. It is perhaps ok to mention the complex in this way as an abbreviation through-out the text, but in this section of the paper please explicitly write the exact chemical formula of the compound (ie. [Cu(Tpy)₂]Cl₂·xH₂O).

Pg 6. Preparation of Cu(Tpy)₂@C is detailed. Cu(Tpy)₂@”glassy carbon” is missing. Please add the missing description since there is a critical comparison in the electrocatalytic section of the paper.

Response: We are grateful to the reviewer’s constructive comments which have led to a significant improvement of our manuscript. Point-to-point responses are provided for both the general and detailed comments as follows.

Point-to-point Response to Reviewer #3:

Comment 1: To the best of our knowledge, this is the first example implementing Cu(Tpy)₂²⁺ to promote electroreduction of CO₂, which makes this article both exciting and questionable. The octahedrally coordinative complex, Cu(Tpy)₂²⁺ does not seem

well-suited to promote catalytic transformations, yet the authors are clearly observing high catalytic activity in the experimental data presented. This is fundamentally different from other example-comparisons which have open coordination sites to bind CO₂, such as CoTPP, CoPCc, NiPCc, etc. Operando Raman and FTIR performed by the authors here suggests that the ligand is hemilabile and non-innocent leading to the baton-like mechanism. The authors' explanation is reasonable given the data and calculations at hand, but leaves the reader wondering why is Cu(Tpy)₂²⁺ not catalytically active as a homogenous species, and why is Cu(Tpy)₂²⁺ far less active on glassy carbon. The authors provide some discussion about the effects of using ketjan black, and these are somewhat different reasons than the literature precedence for multiwalled carbon nanotubes. There are lingering questions such as (1) what hinders the proposed mechanism, especially in the homogenous phase (ie. in MeCN, DMF, DCM, etc.) where lability of the ligand is likely to take place,

Response 1: Thanks for your constructive comments. The molecular catalysts based on metal (Fe, Co, Ni, etc.) porphyrins, phthalocyanine and related derivatives with open coordination sites to bind CO₂ have been widely recognized in CO₂RR studies. These molecular catalysts, either as homogeneous catalysts in the solution or heterogeneous catalysts supported by carbon materials, have been shown effective for CO₂ conversion. (*Adv. Energy Mater.* 2019, 9, 1900090; *Energy Environ. Sci.*, 2020, 13, 374--403). Though less popular, polypyridine metal complexes have also been used in electrochemical CO₂RR, as they could not only generate stable well-defined complexes but also allow for the storage of multiple reducing equivalents across the entire molecule (*Chem. Soc. Rev.*, 2017, 46, 761-796; *Inorg. Chem.* 2015, 54, 12002-12018). As a result, they have been frequently studied in the context of CO₂ electroreduction in organic solvents, most often acetonitrile (CH₃CN) or N,N-dimethylformamide (DMF). (*Phys. Chem. Chem. Phys.*, 2014, 16, 13635; *Chem. Rev.* 2018, 118, 4631-4701). Polypyridyl complexes based on scarce and pricy metals, such as Re, Ru and Rh, were examined in very early studies (*Inorg. Chem.*, 1988, 27, 4582-4587; *Inorg. Chem.*, 1991, 30, 86-91), while recent efforts have been mostly focused on more abundant and cheaper first-row transition metals, such as Mn, Co and Ni (*ACS Catal.* 2020, 10, 1961-

1968; *Chem. Commun.*, 2014, 50, 1491; *Phys. Chem. Chem. Phys.*, 2014, 16, 13635; *J. Am. Chem. Soc.* 2021, 143, 3764), as electrocatalysts for the reduction of CO₂ to CO. However, reports on the octahedrally coordinative Cu-polypyridyl complexes as catalysts for CO₂ reduction are rare. The cyclic voltammograms of Cu-polypyridyl complexes in organic solvents reported by Durand and Fontecave *et al.* showed cathodic current enhancement under CO₂, which was, however, accompanied by deposition behavior on glassy carbon electrodes arising from the poor electronic transport of Cu-polypyridyl complexes (*Electrochim. Acta*, 1988, 33, 581-583; *Phys. Chem. Chem. Phys.*, 2014, 16, 13635).

As shown in **Fig. N7a**, the Cu(Tpy)₂²⁺ still showed cathodic current response just below 5 mA cm⁻² at -1.0 V vs. RHE under CO₂. Indeed, Cu(Tpy)₂²⁺ was less catalytically active as a homogenous species when compared with Cu(Tpy)₂@C in a CO₂-saturated 0.5 M KHCO₃ electrolyte, which was mostly likely because of its poor electrical conductivity and sluggish catalytic kinetics. That molecular catalysts failed short in electron transport and required supporting/immobilizing network such as carbon to achieve desired catalytic activities has been widely recognized in CO₂RR studies, i.e., using CoPc, CoTPP, CoPPc molecules, etc. (*Chem* 2017, 3, 652, *Angew. Chem.* 2017, 129, 1, *Nat. Commun.* 2018, 9, 2671). The present work adopted Ketjen black as a low-cost electron-conducting network to noncovalently immobilize Cu(Tpy)₂ thereby offering well-defined catalytic sites and sufficient electronic conduction. This immobilization strategy enhanced the catalytic activity of Cu(Tpy)₂ by promoting electron transfer and enabling a higher utilization of catalytic sites. Meanwhile, the uniform molecular-level dispersion of Cu(Tpy)₂ in the carbon network effectively prevented any possible deactivations induced by aggregation or dimerization. All these merits contributed to the high catalytic activity presented by the Cu(Tpy)₂@C electrocatalyst in this work.

Figure N7 CO₂RR performance of Cu(Tpy)₂ and Cu(Tpy)₂@C electrocatalysts. (a) Linear sweep voltammetry of neat Cu(Tpy)₂ and Cu(Tpy)₂@C in a CO₂-saturated 0.5 M KHCO₃ electrolyte. (b) FEs for CO and H₂ production of neat C, Tpy, Cu(Tpy)₂ and Cu(Tpy)₂@C. (c) FE(CO)s of Cu(Tpy)₂ and Cu(Tpy)₂@C at different applied potentials. (d) TOFs calculated for Cu(Tpy)₂@C at different potentials. (e) Chronoamperometry and FEs for CO formation at a fixed potential of -0.6 V vs. RHE for Cu(Tpy)₂@C.

Regarding the hinderance to demonstrate the present mechanism in the homogenous phases (i.e., MeCN, DMF, DCM, etc), we explored, as an typical example, the scenario of Cu(Tpy)₂²⁺ under CO₂ in DMF/H₂O (95:5, v:v) with 0.1 M of TBAP as supporting electrolyte. As shown in Fig. N8a, Cu(Tpy)₂²⁺ also afforded a cathodic current response slightly higher than that in a KHCO₃ electrolyte (Fig. 7a). Meanwhile,

the Cu(Tpy)_2^{2+} showed good CO_2 -to- CO selectivity with a high $\text{FE}(\text{CO})$ of 95.5 % at -0.6 V vs. RHE (**Fig. N8b**). However, as shown in **Fig. N9**, during the chronoamperometry test performed at -0.6 V vs. RHE in the organic electrolyte system, both the current density and $\text{FE}(\text{CO})$ gradually decreased, suggesting the deactivation of Cu(Tpy)_2^{2+} .

Fig. N8 (a) LSV of Cu(Tpy)_2 under Ar (black) and CO_2 (red) atmospheres. (b) FEs of Cu(Tpy)_2 at different applied potentials. The working electrode use was a glassy carbon electrode, the counter electrode was a platinum wire, and the reference electrode was a Ag/AgCl. The solvent system used was DMF/ H_2O (95:5, v:v), with 0.1 M of TBAP as supporting electrolyte.

Fig. N9 Chronoamperometry and FEs for CO and H_2 formation at a fixed potential of -0.6 V vs. RHE for Cu(Tpy)_2 under CO_2 atmospheres in DMF/ H_2O (95:5, v:v), with 0.1M of TBAP.

To summarize, we considered the homogeneous phases unsuitable for understanding the CO₂-to-CO reaction mechanisms of Cu(Tpy)₂²⁺, due to some intrinsic disadvantages of homogeneous molecular catalysts. For example, (1) the molecules could only be dissolved in organic solvents like MeCN, DMF, DCM, which was non-economical and non-environmentally friendly, and more importantly, seriously hinder the detection of intermediates for CO₂RR using *in-situ/operando* spectroscopic techniques, especially Raman techniques due to the interference from the electrolyte signals. (2) Only molecules close to the electrode in the diffusion layer could effectively take part in the reaction process, thereby lowering the utilization of molecules and limiting *in-situ* characterization testing lines. (3) Finally, some molecules or their corresponding intermediates tend to dimerize or aggregate during the reaction process, which results in the deactivation of catalysts. After all, it is difficult to elucidate the reaction mechanism during an unstable electrolysis process, i.e., as shown in **Fig. N9**.

It should be noted that **Fig. N7a** shown above corresponds to **Fig. 2a** in the revised manuscript.

Comment 2: *and (2) is the proposed mechanism specific to the carbon surface or does it also operate on other carbon supports? Altogether this discussion, or lack thereof, may not disqualify the article from publication, but it would certainly represent value-added for a journal with such a broad readership. Comments below should be addressed before reconsidering publication.*

Response 2: This is a valuable suggestion. According to the reviewer's suggestion, the Cu(Tpy)₂ hybridized with other forms of nano-carbons including multi wall carbon nanotubes (MWCNTs), single-wall carbon nanotubes (CNTs), Vulcan carbon-72R (XC) were studied (**Fig. N1c,d**). Inductively coupled plasma mass spectrometry (ICP-MS) was employed to determine the Cu amount and to derive the Cu(Tpy)₂ content in the hybrid materials. The Cu content is 0.260 wt.% of Cu(Tpy)₂@C, similar to those of Cu(Tpy)₂@MWCNTs (0.258 wt.%), Cu(Tpy)₂@CNTs (0.263 wt.%) and Cu(Tpy)₂@XC (0.262 wt.%).

Fig. N1 (a) LSV and (b) FEs(CO) acquired using various alkali bicarbonate CO₂-saturated aqueous solutions. (c) LSV and (d) FE(CO)s at -0.6 V vs. RHE of Cu(Tpy)₂ catalysts immobilized on different carbon supports including Ketjen black (KB), multi wall carbon nanotubes (MWCNTs), single-wall carbon nanotubes (CNTs), and Vulcan carbon-72R (XC) using a CO₂-saturated 0.5 M KHCO₃ electrolyte.

As shown in **Fig. N1c**, Cu(Tpy)₂@KB (adopted in the manuscript), Cu(Tpy)₂@MWCNTs, and Cu(Tpy)₂@CNTs showed identical current density responses. Cu(Tpy)₂@XC afforded lower current densities, likely due to the lower graphitic degree of XC, and thus weaker π - π interactions with Cu(Tpy)₂ and slower electron conduction when compared with KB, MWCNTs and CNTs (*Nat. Commun.*, 2017, 8, 14675). Furthermore, FEs(CO) at -0.6 V vs. RHE of Cu(Tpy)₂ immobilized on different carbon supports were also acquired, with no significant discrepancy (**Fig. N1d**). Seemingly, all carbon supports could offer well-defined catalytic sites and sufficient electronic conduction, thereby delivering similar electrolytic performances. Therefore, the selection of carbon supports should not affect our mechanism study and thus the main findings of this work. To clarify, a few sentences were provided now on **Page 3**: “*Supplementary Fig. S5c-d compare the electrocatalytic performances of Cu(Tpy)₂ catalysts immobilized on four different carbon supports as shown in. Clearly, the carbon support type exerts little influence on the catalytic activity and CO selectivity.*”

Therefore, we selected $\text{Cu}(\text{Tpy})_2@C$ with the Ketjen black support for further study considering its slightly higher $\text{FE}(\text{CO})$ output.”

To be more cautious, as shown in **Fig. N10**, we studied the catalytic mechanism on other carbon supports (such as MWCNTs). The peaks at 247 and 354 cm^{-1} could be assigned to the Cu-N bonding environment in the hybrid and *CO. The Cu-N peak clearly shifted to the right, during which the augment of *CO intermediates peaks occurred. All these key observations were consistent with those for $\text{Cu}(\text{Tpy})_2@C$ in our manuscript..

It should be noted that **Fig. N1 (c, d)** is provided as **Fig. S5 (c, d)** in the revised Supplementary Information.

Fig. N10 Time-dependent *operando* Raman spectra of $\text{Cu}(\text{Tpy})_2@MWCNTs$ under electrocatalytic conditions.

Comment 3: Pg 1, column 2. “understanding” is misspelt (understaidng).

Response 3: Thanks for the reviewer’s careful suggestion. According to your question, we have already revised the typo. Detailed revises are provided in the following.

Pg 1, column 2. “understaidng” has been changed into “*understanding*”

Comment 4: *Figure 1a. The authors depict $\text{Cu}(\text{Tpy})_2^{2+}$ complex, a distorted octahedral metal complex, in a highly irregular geometry which strains the Tpy ligands to be coplanar with the hexagonal graphene-like structure. Is the π - π interaction between Tpy and the carbon surface large enough to induce such a distortion? The depiction is never-the-less misleading because it is inconsistent with the proposed mechanism (Figure 1a, Figure 4) involving the octahedral configuration.*

Response 4: Thanks! The $\text{Cu}(\text{Tpy})_2^{2+}$ complex is a distorted octahedral metal complex. However, the $\text{Cu}(\text{Tpy})_2^{2+}$ complex immobilized on the Ketjan black ($\text{Cu}(\text{Tpy})_2@C$) in **Figure 1a** is just a schematic drawing of $\text{Cu}(\text{Tpy})_2@C$. Sorry for bring you brouble! In order to embody a distorted octahedral metal complex of $\text{Cu}(\text{Tpy})_2^{2+}$, the schematic drawing of $\text{Cu}(\text{Tpy})_2@C$ was redrawn (**Fig. N11**). Even if $\text{Cu}(\text{Tpy})_2^{2+}$ complex was fixed on the carbon surface by π - π interactions, it remained the original structure according to the characterizations of Raman spectroscopy, ultraviolet-visible spectroscopy (UV-vis), and X-ray photoelectron spectroscopy (XPS) in the original **Fig.1(b-d)**.

Fig. N11 Enlarged view of the schematic drawing of $\text{Cu}(\text{Tpy})_2^{2+}@C$.

Comment 5: *Citation 28 is sometimes inappropriately used as a reference for the geometric parameters regarding $\text{Cu}(\text{Tpy})_2^{2+}$. Please consider including <https://doi.org/10.1021/ic50180a040> as the corrected citation.*

Response 5: We are very grateful for the reviewer’s comment. We have replaced the

reference [28] accordingly: "...Cu²⁺ is bonded to six N atoms in a strongly distorted octahedral coordination."²⁸".

28 Allmann R., Henke W. & Reinen D. Presence of a static Jahn-Teller distortion in Copper(II) terpyridine complexes. 1. crystal structure of Cu(terpy)₂(NO₃)₂, *Inorg. Chem.*, **17**, 378-382 (1978).

Comment 6: Pg 2 column 2. The authors introduce wettability experiments, however it is not clear based on the written text or Figure S3 captions what surface Cu(Tpy)₂@C is being compared to (i.e., denoted in text as Cu(Tpy)₂). May the authors clarify what the surface materials are, and consider the time dependence on the contact angle measurement. This last point is particularly important in the case of porous materials which may induce changes to contact angles over time (<https://doi.org/10.1016/j.carbon.2015.01.041>). Notably, the gas isotherm data (Figure S4) does corroborate the conclusions based also on the contact angle data (Figure S3).

Response 6: We are very grateful for the reviewer's comment. Ketjan black (KB) was considered superior to other carbon blacks because of its open pore structure and high electrical conductivity (*J. Power Sources* 2017, 360, 383e390). So, KB was used to noncovalently immobilize Cu(Tpy)₂ forming heterogeneous molecular catalysts (hereinafter referred to as Cu(Tpy)₂@C) thereby not only potentially offering well-defined catalytic sites and sufficient electronic conduction, but enhanced surface contact ability (*Carbon*, 2015, 87, 44-60). As for Cu(Tpy)₂@C, the surface materials contained KB with mesoporous structure and evenly dispersed Cu(Tpy)₂ molecular on KB, which greatly increased the contact area, thus providing more reactants adsorption sites. As a comparison, on the Cu(Tpy)₂ molecular surface, Cu(Tpy)₂ molecules stacked on each other, hindering the adsorption of reactants.

Fig. N12 Contact angles measured for Cu(Tpy)₂ and Cu(Tpy)₂@C electrodes with water drops on top.

Bearing this in mind, the contact angles were measured for Cu(Tpy)₂, C and Cu(Tpy)₂@C electrodes in the water droplet experiments (**Fig. N12**). Cu(Tpy)₂@C and C showed smaller contact angles than Cu(Tpy)₂, which could be attributed to the hydrophilic surface of Cu(Tpy)₂@C and the porous structure of the KB. Furthermore, the rapid up-take of water on the Cu(Tpy)₂@C surface suggested that Cu(Tpy)₂@C was more hydrophilic than Cu(Tpy)₂ and C. The hydrophilic Cu(Tpy)₂@C electrode could afford a larger electrochemical active area and a lower adhesion forced to a gas bubble (i.e., of CO) when compared with Cu(Tpy)₂. This not only allowed fast charge transfer reaction occurring at small overpotentials and large currents, but also led to a quick release of CO gas bubbles from the electrode surface to ensure intimate catalyst/electrolyte contact.

Fig. N13 CO₂ adsorption isotherms of neat Cu(Tpy)₂, C and Cu(Tpy)₂@C at 25 °C.

Before we move to the isotherms, we would like to apologize for our mistakes of mislabeling C and Cu Cu(Tpy)₂ in **Supplementary Fig. S4**, which we have corrected and updated in the revised version (also copied above). **Fig. N13** showed the CO₂ adsorption isotherms of neat Cu(Tpy)₂, C and Cu(Tpy)₂@C measured at 25 °C. Among them, the CO₂ adsorption capacity of KB was higher than that of the Cu(Tpy)₂ molecular because of the mesoporous structure of KB. A higher CO₂ adsorption capacity of Cu(Tpy)₂@C was not only ascribed to the presence of KB with a mesoporous structure but also well-defined catalytic sites of Cu(Tpy)₂ on KB to uptake CO₂ molecules. The strong adsorption capacity for CO₂ improved the local concentration of CO₂ and accelerated mass transport on the surface of catalysts, facilitating the electroreduction of CO₂ to CO product.

It should be noted that **Fig. N12** and **Fig. N13** are provided as **Fig. S3** and **Fig. S4** in the revised Supplementary Information.

Comment 7: *Figure S5 right. There are three signals in the GC-TCD. The signal for H₂ is labeled, but there is no reference to what eluate produces the other two signals: N₂, O₂, CO, etc.?*

Response 7: Thanks for pointing out this issue. The other two peaks are O₂ and N₂, respectively (now labeled in the updated **Fig. N14**). O₂ and N₂ peak appeared because of a small amount of air into the gas chromatography during CO₂RR.

Accordingly, we have added **Fig. N14** as **Fig. S6** in the revised Supplementary Information.

Fig. 14 Gas chromatograms profiles of CO and H₂. (a) and (b) are respectively FID and TCD signals captured on Cu(Tpy)₂@C electrode at an overpotential of -0.40 V vs. RHE in a 0.5 M CO₂-saturated KHCO₃ aqueous solution.

Comment 8: Pg 2-3. The authors report significant improvement in electrocatalytic activity by introducing Ketjan black as the carbon support for Cu(Tpy)₂, and this is well supported by the electrolysis data. It is rather interesting that the Tafel analysis reveals different Tafel slopes when comparing Cu(Tpy)₂ on ketjan black versus glassy carbon (Figure S7). The authors state that the much larger Tafel slope with ketjan black is caused by faster reaction kinetics, and this is likely the case. However, Tafel slopes bare consequential meaning pertaining to the mechanism. Tafel slopes ranging from 134-167 mV dec⁻¹ have literature precedence for CO₂ electroreduction. (<https://doi.org/10.1021/acscatal.8b02181>). A much higher Tafel slope, as reported by the authors here, appears to be unique. Therefore, this finding warrants further discussion because there appears to be mechanistic differences by simple virtue of the carbon support.

Response 8: We appreciate this valuable comment. Tafel analysis was widely used to determine the reaction mechanisms, but with certain limitations. Normally, Tafel slope was determined through the dynamic control of the reaction rate under sufficiently low

overpotential, while the observed Tafel slope would be convoluted by mass transport limitations (*ACS Catal.* 2018, 8, 8121-8129). In our manuscript, the Cu(Tpy)₂ on glassy carbon revealed a much larger Tafel slope than Cu(Tpy)₂@C, which was most probably owing to the aggregation of Cu(Tpy)₂ molecules during the reaction process, resulting in transport limitations (*ACS Energy Lett.* 2018, 3, 1381-1386). To be cautious, we measured the Tafel slopes at different Cu(Tpy)₂ loadings, and the relationship was shown in **Fig. N15**. An obvious “saturation” of Tafel slope was reached at the loading of 100%, and the peak value was very close to 254 mV/dec. Therefore, the value of 254 mV/dec was ascribed to the transport limitations of Cu(Tpy)₂ on the glassy carbon. Additionally, the measured Tafel slopes for CO₂RR on many molecular catalysts distribute widely from ~120 mV/dec to >500 mV/dec due to the such transport limitations (*Coordin. Chem. Rev.* 2020, 422, 213435; *Nat. Commun.* 2020, 11, 497). The decrease in Tafel slope at the Cu(Tpy)₂@C was attributed to the uniform molecular-level dispersion of Cu(Tpy)₂ molecules in the electron-conducting carbon network, which alleviated transport limitations and improved the reaction kinetics. Based on this, through simple virtue of the carbon support do not cause the difference of reaction pathway.

Fig. N15 Tafel slope of Cu(Tpy)₂@C as a function of loading.

Comment 9: *There is no description in the SI or elsewhere describing the conditions used to collect EIS (ie. voltage, amplitude, frequency, electrolyte, etc). Was it collected at relevant potentials for CO₂ electroreduction? Perhaps at the open circuit potential? A Bode plot alongside the Nyquist plot would also be helpful, and a full Nyquist plot of Cu(Tpy)₂ on glassy carbon showing far end behavior at high impedance could glean important information.*

Response 9: We are very grateful for the reviewer’s comment. The EIS spectroscopy measurement was carried out by applying an AC voltage with 5 mV amplitude in a frequency range from 1000 Hz to 0.1 Hz at overpotential of -0.6 V (vs. RHE) in a CO₂-saturated 0.5 M KHCO₃ electrolyte. The EIS was collected at relevant potentials for CO₂ electroreduction. Detailed interpretation was provided now on **Page 4** of the Supplementary Information. *“The EIS spectroscopy measurement was carried out by applying an AC voltage with 5 mV amplitude in a frequency range from 1000 Hz to 0.1 Hz at overpotential of -0.6 V (vs. RHE) in a CO₂-saturated 0.5 M KHCO₃ electrolyte.”*

Fig. N16 EIS data in terms of (a) Nyquist and (b) Bode plots of Cu(Tpy)₂ and Cu(Tpy)₂@C measured using a CO₂-saturated 0.5 M KHCO₃ electrolyte; the inset is the equivalent circuit used to simulated the experimental impedance data.

Meanwhile, according to the review’s suggestion, Nyquist plots along with Bode plots were provided for both Cu(Tpy)₂ and Cu(Tpy)₂@C as updated in **Fig. N16** (also copied above). The impedance spectra were fitted using the equivalent circuit shown in the inset of **Fig. N16a**. We agree at high impedance range useful information related to, i.e., diffusion, might be extracted from Nyquist plots. Unfortunately, even at the lowest frequency of 0.1 Hz, no Warburg diffusion behavior has yet initiated. Nonetheless, the huge difference between the charge transfer resistances of Cu(Tpy)₂ and Cu(Tpy)₂@C

suffices to claim the kinetic superiority of Cu(Tpy)₂@C.

It should be noted that we have added **Fig. N16** as **Fig. S9** in the revised Supplementary Information.

Comment 10: *Pg 4-5, Figure 3a. The formation of N-COOH intermediate species is based on a small shift in the Cu-N signal at 247 cm⁻¹, which is also rather broad. Could this shift be an artifact of identifying the peak of this broad signal? May the authors convince the reader that this assignment of the shifting peak is real? Perhaps consider summing a weighted average of the calculated signals shown in Figure 3g.*

Response 10: We are very grateful for the reviewer's comment. Based on the reviewer's questions, in order to verify the repeatability and veracity of the Raman data. *operando* Raman spectroscopy analysis was conducted in a short period of time. The Raman spectra were collected in every 0.1 seconds from 0-2.5 s under a static scanning mode at a controlled potential of -0.45 V vs. RHE (This data has been completed three times). As shown in **Fig. N4a-d**, the peaks at 247, 354, and 1630 cm⁻¹ could be assigned to the Cu-N bonding environment in the hybrid, *CO, and *COOH, respectively. One important takeaway from **Fig. 4a-d** was that the Cu-N peak also clearly shifted to the right, during which the augment of *COOH and *CO intermediates peaks occurred. We favorably linked this continuous Cu-N peak shift to the (i) chemical adsorption of CO₂ to the N site that is coordinated to the center Cu atom of Cu(Tpy)₂@C, and (ii) subsequent breaking of Cu-N bonds. Such Cu-N Raman peak shift resulting from bonding environment changes,^{48,49} have been widely reported in the literature. In addition, it is worth mentioning that the Raman signals of the adsorption are discrete lines in theoretical calculations, which were expanded in Gaussian functions to make them comparable with the experimental results.

Figure N4 | Time-dependent *operando* Raman spectra of Cu(Tpy)₂ under electrocatalytic conditions. The spectra were measured using (a-d) a CO₂-saturated 0.5 M KHCO₃ solution and (e-h) an Ar-saturated 0.5 M KHCO₃ solution, respectively. The potential was controlled at -0.45 V vs. RHE. The results are displayed in three scanning subsections, namely (a, e) 220-270 cm⁻¹, (b, f) 280-400 cm⁻¹, (c, g) 1040-1120 cm⁻¹, and (d, h) 1610-1640 cm⁻¹. (i) Comparisons among the DFT-calculated and operando Raman spectra of Cu(Tpy)₂. The experimental line was extracted from the top spectra in panels a-d at the end of 2.5 s.”

Comment 11: Figure S11a shows rather irrefutably that an -COOH species is formed during electrolysis. In Table S5 the Cu-COOH species is shown to be unfavored, even though this bonding mode is often invoked for numerous metal-based catalysts. Alternatively, the bonding mode might be “Cu-OOHC-N as in

<https://doi.org/10.1021/ja039577h>. Please comment on this alternative interpretation.

Response 11: At the initial stage, the bridging absorption pattern as suggested by the referee has been taken into consideration in the calculation. Unfortunately, it could not be located and would transfer into the N-COOH absorption pattern as the local minimum along the optimized process. This is possibly due to the large ring strain in the Cu-O-C(OH)-N four-member ring of the bridge-like adsorption pattern. We have added this sentence in the main text (**Page 5**). *“It is worth mentioning that we carefully checked all possible adsorption patterns of COOH species, especially the bridge-like model⁵⁷ as Cu-OOHC-N; however, it could not be formed in this system most likely owing to the large ring strain.”*

Comment 12: Pg 6. *“Preparation of the Cu(Tpy)₂.”* Until now the authors refer to the copper(II) complex as Cu(Tpy)₂, but actually the complex is cationic with two chlorides to balance the charge. It is perhaps ok to mention the complex in this way as an abbreviation through-out the text, but in this section of the paper please explicitly write the exact chemical formula of the compound (ie. [Cu(Tpy)₂]Cl₂•xH₂O).

Response 12: Thanks for pointing out this issue. As suggested, *“Preparation of the Cu(Tpy)₂.”* has been changed into *“Preparation of the [Cu(Tpy)₂]Cl₂•xH₂O”* wherever it applies.

Comment 13: Pg 6. *Preparation of Cu(Tpy)₂@C is detailed. Cu(Tpy)₂@“glassy carbon” is missing. Please add the missing description since there is a critical comparison in the electrocatalytic section of the paper.*

Response 13: Thanks for the reviewer’s suggestion. We have added the description of Cu(Tpy)₂@glassy carbon in the Methods part (also copied below).

“Preparation of the Cu(Tpy)₂@glassy carbon: 3 mg [Cu(Tpy)₂]Cl₂•xH₂O was sonicated for 2 h in 1.0 mL ethanol and 0.1 mL Nafion® 117 solution. Subsequently, 3 μL catalyst ink was applied onto a glassy carbon electrode and allowed to dry in air, achieving a catalyst loading of 0.13 mg cm⁻². A CO₂-saturated electrolyte was prepared by passing CO₂ into 0.5 M KHCO₃ aqueous solution for 30 min.”

REVIEWER COMMENTS

Reviewer #2 (Remarks to the Author):

The implemented additional Raman measurements and improved mechanistic discussions have clearly answered my previous questions. The results presented in this manuscript will help fellow researchers deepen their understanding in CO₂RR over molecular catalysts on a more general scale.

I recommend publication of this manuscript after fixing some small errors. E. g. direction of the light cone in the scheme of fig. 1A is incorrect (What an objective does is to focus light in a small area, not the reverse)

Please check the manuscript carefully once again.

Reviewer #3 (Remarks to the Author):

Please view attachment for response.

Review following rebuttal

Title: Mechanistic insights into CO₂ conversion chemistry of molecular electrocatalyst using accessible operando spectrochemistry

Authors: Zhang, H.; Xu, C., Zhan, X.; Yu, Y.; Zhang, K.; Luo, Q.; Gao, S.; Yang, J.; Xie, Y.

Journal: *Nature Communication*

Response:

In all, the authors have addressed the major comments during revision, and the paper has strengthened significantly. Principally, the major concern of stability of Cu(Tpy)₂@C as the active catalyst species has been further supported by HAADF-STEM, and this data alongside the pre-existing electrolysis, FT-IR and XPS data has unequivocally indicated that the supported-catalyst at the start of the experiment remains intact. I would still maintain that deploying Cu(Tpy)₂ is an unlikely candidate for such novel and selective CO₂-to-CO conversion. As the authors are aware, Cu(Tpy)₂ in the organic phase is unstable as a CO₂RR catalyst and results in deposition of Cu containing material onto the carbon electrode surface as noted by Artero and coworkers: "Evidence for deposition behavior on a glassy carbon electrode under CO₂ was observed during cyclic voltammetry experiments. . ." (*Phys. Chem. Chem. Phys.*, **2014**, 16, 13635). In the follow up work here, the authors have indirectly observed the outcome of CO₂RR with Cu(Tpy)₂ in DMF/H₂O solution in the form of a product distribution containing H₂, CO, and HCOOH which likely ensues because of degradation leading to reactive Cu-hydride species, and in my opinion should be included in the paper revision as evidence for their system's stability. While discussion of Cu(Tpy)₂ as a catalyst in these various environments currently appears outside the scope of the paper, and that seems to be ok, the relative stability of Cu(Tpy)₂ in a distorted geometry as a supported-molecular catalyst may inform future catalyst design. This inclusion would require only a brief mention in the main text and would most likely draw even more attention and citations to the paper.

Most common during the review process, notation can easily go astray. I suggest only three major edits regarding readability that would lead to an incontrovertible recommendation for publication. The minor (and very minor) comments appearing below can be considered as a matter of personal preference.

Major comments:

- A complication that impacts the paper globally and has been highlighted in the caption of Figure 1 and 2:

"physicochemical characteristics of Cu(Tpy)₂ and Cu(Tpy)₂@C."

"(b) FEs for CO and H₂ production of neat C, Tpy, Cu(Tpy)₂ and Cu(Tpy)₂@C."

Cu(Tpy)₂@C has clear notation referring to the molecular catalyst-support system denoted as catalyst@support, however in some instances the readability becomes quite low. Does Tpy and Cu(Tpy)₂ indicate dissolved species in aqueous electrolyte? If so, the concentrations of these compounds in solution should be listed in the figure caption and possibly elsewhere too. If not, please consider referring to it as, for example, Tpy@support. “@KB” and “@GC” could denote the two surfaces of interest with clarity such as in the SI with designations (*ie.* Cu(Tpy)₂@KB, MWNCTs, CNTs, XC).

- Might the authors be more specific in the title? For example replacing “molecular catalyst” with “Cu(Tpy)₂” would be appropriate change given the scope of the paper.
- The preparation of electrodes that include glassy carbon is often trivialized and seems a bit lacking in description here as well. What size glassy carbon plates and who is the manufacturer? A common issue in electrode fabrication of this type is covering blank glassy carbon with insulating material near the edges of a plate as to prevent contact with electrolyte solution. In short, the readership would benefit from a detailed description of the fabrication process for all catalyst@support.

Minor comments:

- Consider drawing attention to the novelty of Cu(Tpy)₂ as a non-obvious choice for a supported-molecular catalyst. Ni(Tpy)₂ or Co(Tpy)₂ would have been a more obvious choice of metal complex provided the literature precedence for these complexes to promote CO₂-to-CO conversion. The stability and electrocatalysis of Cu(Tpy)₂ in a strained geometry on the carbon surface is very novel.
- Is the immobilization of Cu(Tpy)₂ on carbon reversible? π - π stacking is typically a weak physisorption interaction that can be disrupted by a select-solvent. In other words, what does the homogenous phase Raman spectrum of Cu(Tpy)₂ in DMF solutions before and after electrolysis tell us about the structural changes (if any)? Does the act of mixing Cu(Tpy)₂ with carbon result in an irreversible structural change to Cu(Tpy)₂?

Very minor comment:

- Pg 1. Introduction. The authors’ state: “Compared with oxygen evolution reaction and ORR, CO₂RR is a more complicated process involving proton coupled multi-electron transfer.” – I agree the CO₂RR is more complicated, but all reaction pathways listed involve proton coupled electron transfer mechanisms. The complexity in CO₂RR is typically due to the multiple pathways leading to multiple products, thus selectivity is often the most challenging aspect with heterogenous CO₂RR catalysis. Your article addresses the challenge very eloquently by deploying a supported-molecular catalyst!

Responses to the Reviewers' comments and a summary of the changes made to the manuscript: NCOMMS-22-03394-T.

Many thanks to the Reviewers for having given us valuable suggestions and comments to improve the quality of the manuscript.

We have carefully read the Reviewers' comments to further improve this manuscript. And we acknowledge the Reviewers' positive comments, i.e., "The implemented additional Raman measurements and **improved mechanistic discussions have clearly answered my previous questions**. The results presented in this manuscript will **help fellow researchers deepen their understanding in CO₂RR** over molecular catalysts on a more general scale.", "I **recommend publication of this manuscript** after fixing some small errors", "the authors have **addressed the major comments during revision**, and the paper has **strengthened significantly**.", "The complexity in CO₂RR is typically due to the multiple pathways leading to multiple products, thus selectivity is often the most challenging aspect with heterogenous CO₂RR catalysis. **Your article addresses the challenge very eloquently by deploying a supported-molecular catalyst**". "I suggest only three major edits regarding readability that would lead to an **incontrovertible recommendation for publication**".

According to the Reviewers' valuable comments and suggestions, we have supplemented the pertinent experimental data and carefully modified the manuscript. A list of changes made and the corresponding explanations are presented as follows in a point-to-point manner.

The entire comments from Reviewer #2

*The implemented additional Raman measurements and **improved mechanistic discussions have clearly answered my previous questions**. The results presented in this manuscript will **help fellow researchers deepen their understanding in CO₂RR** over molecular catalysts on a more general scale.*

*I **recommend publication of this manuscript** after fixing some small errors. E. g. direction of the light cone in the scheme of fig. 1A is incorrect (What an objective does is to focus light in a small area, not the reverse)*

Please check the manuscript carefully once again.

Response: We are grateful to your great interest and very positive comments on our present work, and we also greatly appreciate your constructive suggestions guiding the revision of our manuscript. We address the comments and questions as detailed below.

Point-to-point Response to Reviewer #2:

Comment 1: I recommend publication of this manuscript after fixing some small errors. E. g. direction of the light cone in the scheme of fig. 1A is incorrect (What an objective does is to focus light in a small area, not the reverse)

Response 1: Thanks for your valuable comment. As the reviewer suggested, the direction of the light cone in the scheme of Fig. 1A has been modified as shown in Fig. N1.

Fig. N1. A scheme illustrating the operando Raman study on the CO₂RR process of Cu(Tpy)₂-KB@GC electrocatalyst.

Comment 2: Please check the manuscript carefully once again.

Response 2: Thanks for the reviewer's suggestion. We have double checked the manuscript.

The entire comments from Reviewer #3

*In all, the authors have **addressed the major comments during revision**, and the paper has **strengthened significantly**. Principally, the major concern of stability of $\text{Cu}(\text{Tpy})_2@C$ as the active catalyst species has been further supported by HAADF-STEM, and this data alongside the pre-existing electrolysis, FT-IR and XPS data has unequivocally indicated that the supported catalyst at the start of the experiment remains intact. I would still maintain that deploying $\text{Cu}(\text{Tpy})_2$ is an unlikely candidate for such novel and selective CO_2 -to-CO conversion. As the authors are aware, $\text{Cu}(\text{Tpy})_2$ in the organic phase is unstable as a CO_2RR catalyst and results in deposition of Cu containing material onto the carbon electrode surface as noted by Artero and coworkers: “Evidence for deposition behavior on a glassy carbon electrode under CO_2 was observed during cyclic voltammetry experiments.” (*Phys. Chem. Chem. Phys.*, **2014**, 16, 13635). In the follow up work here, the authors have indirectly observed the outcome of CO_2RR with $\text{Cu}(\text{Tpy})_2$ in DMF/ H_2O solution in the form of a product distribution containing H_2 , CO, and HCOOH which likely ensues because of degradation leading to reactive Cu-hydride species, and in my opinion should be included in the paper revision as evidence for their system’s stability. While discussion of $\text{Cu}(\text{Tpy})_2$ as a catalyst in these various environments currently appears outside the scope of the paper, and that seems to be ok, the relative stability of $\text{Cu}(\text{Tpy})_2$ in a distorted geometry as a supported-molecular catalyst may inform future catalyst design. This inclusion would require only a brief mention in the main text and would most likely draw even more attention and citations to the paper.*

*Most common during the review process, notation can easily go astray. I suggest only three major edits regarding readability that would lead to an **incontrovertible recommendation for publication**. The minor (and very minor) comments appearing below can be considered as a matter of personal preference.*

Major comments:

- A complication that impacts the paper globally and has been highlighted in the caption of Figure 1 and 2:*

“physicochemical characteristics of $\text{Cu}(\text{Tpy})_2$ and $\text{Cu}(\text{Tpy})_2@C$.”

“(b) FEs for CO and H₂ production of neat C, Tpy, Cu(Tpy)₂ and Cu(Tpy)₂@C.”
Cu(Tpy)₂@C has clear notation referring to the molecular catalyst-support system denoted as catalyst@support, however in some instances the readability becomes quite low. Does Tpy and Cu(Tpy)₂ indicate dissolved species in aqueous electrolyte? If so, the concentrations of these compounds in solution should be listed in the figure caption and possibly elsewhere too. If not, please consider referring to it as, for example, Tpy@support. “@KB” and “@GC” could denote the two surfaces of interest with clarity such as in the SI with designations (ie. Cu(Tpy)₂@KB, MWNCTs, CNTs, XC).

- Might the authors be more specific in the title? For example replacing “molecular catalyst” with “Cu(Tpy)₂” would be appropriate change given the scope of the paper.*
- The preparation of electrodes that include glassy carbon is often trivialized and seems a bit lacking in description here as well. What size glassy carbon plates and who is the manufacturer? A common issue in electrode fabrication of this type is covering blank glassy carbon with insulating material near the edges of a plate as to prevent contact with electrolyte solution. In short, the readership would benefit from a detailed description of the fabrication process for all catalyst@support.*

Minor comments:

- Consider drawing attention to the novelty of Cu(Tpy)₂ as a non-obvious choice for a supported-molecular catalyst. Ni(Tpy)₂ or Co(Tpy)₂ would have been a more obvious choice of metal complex provided the literature precedence for these complexes to promote CO₂-to-CO conversion. The stability and electrocatalysis of Cu(Tpy)₂ in a strained geometry on the carbon surface is very novel.*
- Is the immobilization of Cu(Tpy)₂ on carbon reversible? π - π stacking is typically a weak physisorption interaction that can be disrupted by a select-solvent. In other words, what does the homogenous phase Raman spectrum of Cu(Tpy)₂ in DMF solutions before and after electrolysis tell us about the structural changes (if any)? Does the act of mixing Cu(Tpy)₂ with carbon result in an irreversible structural change to Cu(Tpy)₂?*

Very minor comment:

• Pg 1. Introduction. The authors' state: "Compared with oxygen evolution reaction and ORR, CO₂RR is a more complicated process involving proton coupled multi-electron transfer." – I agree the CO₂RR is more complicated, but all reaction pathways listed involve proton coupled electron transfer mechanisms. The complexity in CO₂RR is typically due to the multiple pathways leading to multiple products, thus selectivity is often the most challenging aspect with heterogenous CO₂RR catalysis. **Your article addresses the challenge very eloquently by deploying a supported-molecular catalyst!**

Response: We greatly appreciate your positive comments on our work and we are grateful to your constructive suggestions guiding the revision of our manuscript. We have addressed the comments and questions accordingly as detailed below.

Point-to-point Response to Reviewer #3:

Comments 1: *In the follow up work here, the authors have indirectly observed the outcome of CO₂RR with Cu(Tpy)₂ in DMF/H₂O solution in the form of a product distribution containing H₂, CO, and HCOOH which likely ensues because of degradation leading to reactive Cu-hydride species, and in my opinion should be included in the paper revision as evidence for their system's stability*

Response 1: Thanks for your valuable comment. According to your suggestion, the outcome of CO₂RR with Cu(Tpy)₂@GC in DMF/H₂O solution were added in the revised paper. To clarify, a few sentences were provided now on **Page 2**: "Before exploring Cu(Tpy)₂-KB@GC's catalytic activity, we tested the catalytic performance of the Cu(Tpy)₂@GC under CO₂ in DMF/H₂O (95:5, v:v) with 0.1 M of TBAP as supporting electrolyte. As shown in Supplementary Fig. S5, Cu(Tpy)₂@GC afforded a notable cathodic current response and a good CO₂-to-CO selectivity with a high FE(CO) of 95.5 % at -0.6 V vs. RHE. However, during the chronoamperometry test performed at -0.6 V vs. RHE in the organic electrolyte system, both the current density and FE(CO) gradually decreased, suggesting the deactivation of Cu(Tpy)₂@GC (Supplementary Fig. S6)."

It should be noted that **Fig. N2** and **N3** are now provided as **Fig. S5** and **S6** in the

revised Supplementary Information.

Fig. N2 (a) LSV of Cu(Tpy)₂@GC under Ar (black) and CO₂ (red) atmospheres. (b) FEs of Cu(Tpy)₂@GC at different applied potentials. The solvent system was DMF/H₂O (95:5, v:v), with 0.1 M of TBAP as supporting electrolyte.

Fig. N3 Chronoamperometry and FEs for CO and H₂ formation at a fixed potential of -0.6 V vs. RHE for Cu(Tpy)₂@GC under CO₂ atmospheres in DMF/H₂O (95:5, v:v), with 0.1 M of TBAP.

Comments 2: A complication that impacts the paper globally and has been highlighted in the caption of Figure 1 and 2:

“physicochemical characteristics of Cu(Tpy)₂ and Cu(Tpy)₂@C.”

“(b) FEs for CO and H₂ production of neat C, Tpy, Cu(Tpy)₂ and Cu(Tpy)₂@C.”
Cu(Tpy)₂@C has clear notation referring to the molecular catalyst-support system denoted as catalyst@support, however in some instances the readability becomes quite low. Does Tpy and Cu(Tpy)₂ indicate dissolved species in aqueous electrolyte? If so, the concentrations of these compounds in solution should be listed in the figure caption and possibly elsewhere too. If not, please consider referring to it as, for example, Tpy@support. “@KB” and “@GC” could denote the two surfaces of interest with clarity such as in the SI with designations (ie. Cu(Tpy)₂@KB, MWNCTs, CNTs, XC)

Response 2: Thanks for your valuable comment. Tpy and Cu(Tpy)₂ do not indicate dissolved species in aqueous electrolyte, instead they are deposited onto the glassy carbon (GC) electrodes by drop casting. According to your question, we renamed the materials covered in the article to enhance readability.

Cu(Tpy)₂-KB: Cu(Tpy)₂ complex immobilized on Ketjen black (KB).

Cu(Tpy)₂-KB@GC: Cu(Tpy)₂-KB deposited on the glassy carbon (GC) electrode.

Cu(Tpy)₂@GC: Cu(Tpy)₂ deposited on the GC electrode.

Tpy@GC: Tpy deposited on GC electrode.

KB@GC: Ketjen black (KB) deposited on GC electrode.

Accordingly, to be accurate we have rephrased all the descriptions of the material, wherever applicable in the manuscript. Detailed changes are provided as shown below.

Page 1, column 16. “As a proof of concept, we synthesize Cu(Tpy)₂ complex and immobilize it on *Ketjen black* (hereinafter referred to as *Cu(Tpy)₂-KB*) to form a model electrocatalyst capable of kilogram-scale production.”

Page 1, column 19. “After casting the *Cu(Tpy)₂-KB* on the glassy carbon (GC) electrode, the resulting *Cu(Tpy)₂-KB@GC* exhibits ...”

Page 2, Fig 1. “physicochemical characteristics of *Cu(Tpy)₂* and *Cu(Tpy)₂-KB*”

Page 3, Fig 2. “FEs for CO and H₂ production of neat *KB@GC*, *Tpy@GC*, *Cu(Tpy)₂@GC* and *Cu(Tpy)₂-KB@GC*”

We have added the description of catalyst@glassy carbon in the Methods part (also copied below).

“Preparation of the Cu(Tpy)₂-KB@GC, Cu(Tpy)₂@GC, KB@GC, Tpy@GC:
Calculated amount of Cu(Tpy)₂-KB, Cu(Tpy)₂, KB, Tpy was sonicated for 2 h in 1.0 mL ethanol and 0.1 mL Nafion®117 solution, respectively. Subsequently, 3 μL catalyst ink was applied onto a glassy carbon electrode and allowed to dry in air, achieving the catalysts loading of 0.13 mg cm⁻². A CO₂-saturated electrolyte was prepared by passing CO₂ into 0.5 M KHCO₃ aqueous solution for 30 min.”

In addition, “Cu(Tpy)₂@C” has been changed to “Cu(Tpy)₂-KB@GC” in the revised paper.

Comment 3: *Might the authors be more specific in the title? For example replacing “molecular catalyst” with “Cu(Tpy)₂” would be appropriate change given the scope of the paper.*

Response 3: Thanks for the reviewer’s careful suggestion. According to your suggestion, “Mechanistic insights into CO₂ conversion chemistry of molecular electrocatalyst using accessible operando spectrochemistry” has been modified to “Mechanistic insights into CO₂ conversion chemistry of Cu(Tpy)₂ electrocatalyst using accessible operando spectrochemistry”

Comment 4: *The preparation of electrodes that include glassy carbon is often trivialized and seems a bit lacking in description here as well. What size glassy carbon plates and who is the manufacturer? A common issue in electrode fabrication of this type is covering blank glassy carbon with insulating material near the edges of a plate as to prevent contact with electrolyte solution. In short, the readership would benefit from a detailed description of the fabrication process for all catalyst@support.*

Response 4: Thanks for reminding us. The preparation of electrodes that include glassy carbon could be found in the *Electrochemical testing* section on **Page 4** of the Supplementary Information. “a glassy carbon (GC) disk working electrode (0.3 cm diameter, Model CHI 104, CH Instruments Inc., USA). The supporting material of GC is Kel-F. Before the electrocatalytic test, the GC was pretreated as detailed below. The surface of GC was cleaned by polishing with MicroPolish Powder 0.05 micron (CH

Instruments, Inc.). Then the processed GC electrode was placed in an aqueous solution containing 1 mM $K_3Fe(CN)_6$ and 0.1 M KCl, and its cyclic voltammetry curve was observed. If the anodic and cathodic peaks are symmetrical with identical peak current values ($I_{PC}/I_{PA} = 1$), and the peak-to-peak potential difference (ΔE_p) was about 60 mV, then the electrode surface was considered to be well processed, otherwise it needs to be re-polished to reach the requirement. Subsequently, calculated amount of catalyst ink was deposited on the GC electrode (referred to as catalyst@GC).”

Comment 5: *Consider drawing attention to the novelty of $Cu(Tpy)_2$ as a non-obvious choice for a supported-molecular catalyst. $Ni(Tpy)_2$ or $Co(Tpy)_2$ would have been a more obvious choice of metal complex provided the literature precedence for these complexes to promote CO_2 -to- CO conversion. The stability and electrocatalysis of $Cu(Tpy)_2$ in a strained geometry on the carbon surface is very novel.*

Response 5: We are very grateful for the reviewer’s comment. In the manuscript, the $Cu(Tpy)_2$ -KB@GC exhibits a high CO selectivity of 99.5% and a remarkable stability of 80 h at -0.6 V vs. RHE, along with exceptionally high turn-over efficiencies (TOFs), all of which are among the best values reported to date. However, we agree that Ni or Co complex would also be interesting to pursue in the future.

Comment 6: *Is the immobilization of $Cu(Tpy)_2$ on carbon reversible? π - π stacking is typically a weak physisorption interaction that can be disrupted by a select-solvent. In other words, what does the homogenous phase Raman spectrum of $Cu(Tpy)_2$ in DMF solutions before and after electrolysis tell us about the structural changes (if any)? Does the act of mixing $Cu(Tpy)_2$ with carbon result in an irreversible structural change to $Cu(Tpy)_2$?*

Response 6: We are very grateful for the reviewer’s comment. (1) The immobilization of $Cu(Tpy)_2$ on KB is not reversible to the best of our knowledge. The $Cu(Tpy)_2$ can be anchored onto the 3D carbon network formed by Ketjen black (KB) through π - π stacking, as verified by Raman spectroscopy, UV-vis, and XPS of before and after the

immobilization of Cu(Tpy)₂ on KB. The long-term stability of Cu(Tpy)₂-KB@GC during electrolysis (Fig. 2e) and the post-electrolysis FT-IR/XPS analysis (Supplementary Fig. S13) also supports the stability of the immobilized structure.

(2) We have tested Cu(Tpy)₂ under DMF/H₂O homogeneous phase and the electrolysis result suggested the deactivation of Cu(Tpy)₂ (Supplementary Fig. S6). Therefore, we consider the further analysis of Cu(Tpy)₂-KB (i.e., using Raman) in DMF solution neither necessary nor helpful to the main scope of this work.

(3) Indeed, mixing Cu(Tpy)₂ with KB induce immobilization of Cu(Tpy)₂ on KB through a weak physisorption interaction (i.e., π - π stacking). However, after the immobilization, Cu(Tpy)₂ maintained the same characteristic peaks in XPS, UV-vis and Raman analysis, and only small peak shift were observed for Cu(Tpy)₂-KB because of the electronic coupling between Cu(Tpy)₂ and KB (Fig. 1b-d). Therefore, we think this immobilization process did not change the major structure of Cu(Tpy)₂ and thus did not qualify for an irreversible structural change to Cu(Tpy)₂.

Comment 7: *Pg 1. Introduction. The authors' state: "Compared with oxygen evolution reaction and ORR, CO₂RR is a more complicated process involving proton coupled multi-electron transfer." – I agree the CO₂RR is more complicated, but all reaction pathways listed involve proton coupled electron transfer mechanisms. The complexity in CO₂RR is typically due to the multiple pathways leading to multiple products, thus selectivity is often the most challenging aspect with heterogenous CO₂RR catalysis. Your article addresses the challenge very eloquently by deploying a supported-molecular catalyst!*

Response 7: Thanks for your positive comments!